# Joint Optimization for 4D Human-Scene Reconstruction in the Wild

**Zhizheng Liu, Joe Lin, Wayne Wu, Bolei Zhou**
Department of Computer Science, University of California, Los Angeles

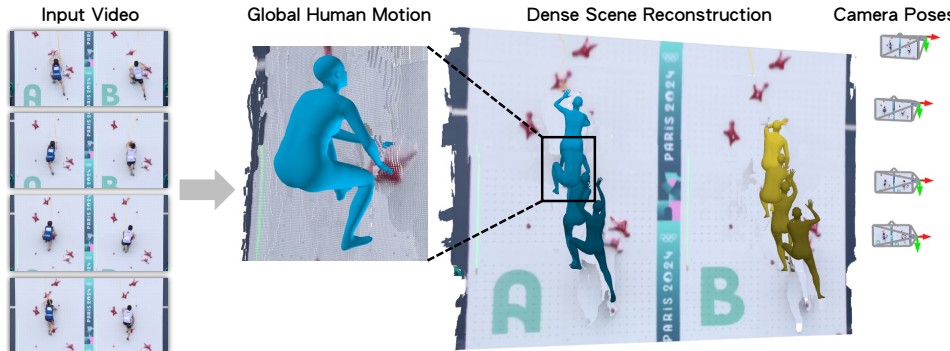

Figure 1: **4D human-scene reconstruction in the wild with JOSH.** Given a web video captured from a single camera as input, JOSH outputs the global human motion, the dense scene reconstruction, and the camera poses with coherent human-scene interaction.

## Abstract

Reconstructing human motion and its surrounding environment is crucial for understanding human-scene interaction and predicting human movements in the scene. While much progress has been made in capturing human-scene interaction in constrained environments, those prior methods can hardly reconstruct the natural and diverse human motion and scene context from web videos. In this work, we propose JOSH, a novel optimization-based method for 4D human-scene reconstruction in the wild from monocular videos. Compared to prior works that perform separate optimization of the human, the camera, and the scene, JOSH leverages the human-scene contact constraints to jointly optimize all parameters in a single stage. Experiment results demonstrate that JOSH significantly improves 4D human-scene reconstruction, global human motion estimation, and dense scene reconstruction by utilizing the joint optimization of scene geometry, human motion, and camera poses. Further studies show that JOSH can enable scalable training of end-to-end global human motion models on extensive web data, highlighting its robustness and generalizability. The code and model are available at https://vail-ucla.github.io/JOSH/.

## 1 Introduction

Humans constantly interact with their surrounding environments: people sit on benches at plazas, walk through crosswalks at intersections, and climb stairs inside buildings. Capturing and understanding human-scene interaction is thus crucial to many applications. For example, analyzing how pedestrians navigate crosswalks and sidewalks is essential for the safe deployment of autonomous driving (Keller & Gavrila, 2013). Meanwhile, designing urban public spaces requires knowing how individuals interact with their surroundings to optimize the group flow, encourage social interaction, and create inviting, accessible environments (Gehl, 2013).

Existing research in human-scene interaction mainly focuses on capturing and synthesizing human motion in pre-scanned 3D scenes (Hassan et al., 2019; Huang et al., 2022; Jiang et al., 2024). They first reconstruct the scene without people in constrained environments with complex sensor setups such as multi-view RGBD cameras and laser scanners and then fit the human motion to the environment. Such complex setups limit data accessibility and the capturing of natural and diverse human-scene interactions in the wild. On the other hand, reconstructing global human motion from

casual web videos has been a rising topic in recent years (Von Marcard et al., 2018; Kaufmann et al., 2023). However, most previous works only reconstruct motion without scene context (Shin et al., 2024; Wang et al., 2024b; Li et al., 2024; Wang et al., 2025b), and the resulting reconstructed motion lacks grounding and meaning from its surrounding environment.

In this paper, we aim to capture human-scene interactions by tackling monocular 4D human-scene reconstruction, which reconstructs the camera pose, the 4D global human motion, and the 3D scene from web videos. Only a few works (Zhao et al., 2024; Luvizon et al., 2023; Liu et al., 2021; Xue et al., 2024) have attempted 4D human-scene reconstruction, by separately performing camera pose estimation, scene reconstruction, and global human motion optimization. However, such approaches ignore the critical interplay between the camera, the human, and the environment, as they can mutually refine each other continuously in a holistic reconstruction. Moreover, as shown in Fig. 3 (a), these approaches can hardly achieve consistency and coherence in detailed human-scene contacts, resulting in physically implausible reconstruction. Many prior works (Shin et al., 2024; Shen et al., 2024; Wang et al., 2024b) also only focus on reconstructing the motion of one single person without ensuring the consistency of multi-human motion in the same world space. Therefore, it is natural and essential to attempt to reconstruct the camera, the scene, and the motions of all humans simultaneously with joint optimization for better accuracy and consistency of 4D human-scene reconstruction.

In this work, we propose JOSH (Joint Optimization of Scene Geometry and Human Motion), a general optimization framework for 4D human-scene reconstruction from monocular videos. As shown in Fig. 1, from a single video, JOSH can simultaneously reconstruct dense 3D environment, camera poses, and 4D global human motion, producing accurate and coherent human-scene interaction data. JOSH initializes the optimization with local human mesh recovery, human-scene contact labels, depth maps, and point correspondences. Instead of performing sequential optimization and reconstruction, JOSH jointly refines the camera poses, the 4D global human motion of all people, and the dense 3D scene point cloud in a single stage, with the key insight that human-scene contact can act as strong constraints that bridge the scene and human motion and guide the joint optimization to produce more precise and consistent 4D human-scene reconstruction results.

JOSH is a general optimization framework, and we experiment with JOSH using different initialization methods. Our results demonstrate that JOSH significantly improves performance in 4D human-scene reconstruction, global human motion estimation, and dense scene reconstruction, compared to approaches that do not apply joint optimization. Noticeably, when initialized with VIMO (Wang et al., 2024b) and MASt3R (Duisterhof et al., 2024), JOSH sets a new state-of-the-art for global human motion estimation, surpassing previous methods by a clear margin. Ablation studies demonstrate the effectiveness of each optimization component in JOSH.

Web videos offer a rich and diverse source of real-world global human motion; however, their unstructured nature makes obtaining reliable ground-truth annotations challenging. Owing to JOSH's strong robustness and generalizability, it can facilitate the scalable training of end-to-end global human motion models by providing accurate pseudo-labels. Further experiments demonstrate that the model trained on web data using JOSH's pseudo-labels significantly outperforms the one trained on datasets with ground-truth labels. This highlights JOSH's effectiveness in enabling scalable training on in-the-wild data. We further design an end-to-end model, JOSH3R (JOint Scene and Human 3D Reconstruction), with a lightweight human trajectory head based on MASt3R Leroy et al. (2024) to predict the relative human transformation directly between two frames, allowing real-time inference as a trade-off to the estimation accuracy.

We summarize our contribution as follows: 1) A general optimization framework, JOSH, is proposed to tackle the challenge of 4D human-scene reconstruction in the wild by jointly optimizing the camera pose, the global human motion, and the scene reconstruction in a single stage. 2) Experiments with different initializations show that JOSH brings significant improvement to 4D human-scene reconstruction, global human motion estimation, and dense scene reconstruction. 3) Further studies show JOSH3R achieves competitive results by training only with labels predicted by JOSH, highlighting JOSH's high potential for scalable training of end-to-end models using extensive web videos.

## 2 PRELIMINARIES

4D human-scene reconstruction involves both global human motion estimation and dense scene reconstruction. We introduce preliminaries of these two tasks in Sec. 2.1 and Sec. 2.2, respectively.

## 2.1 GLOBAL HUMAN MOTION ESTIMATION

In global human motion estimation, the goal is to reconstruct the 4D human motion with SMPL (Loper et al., 2023) human parameters in the world coordinate at every timestep $\{\mathbf{\Theta}_g^t\}_{t=1}^N$. The SMPL model takes the parameters $\mathbf{\Theta}_g^t = \{\boldsymbol{T}_g^t, \boldsymbol{\theta}^t, \boldsymbol{\beta}^t\}$ and outputs the human body mesh vertices $\boldsymbol{V}_g^t \in \mathbb{R}^{6890 \times 3}$, where $\boldsymbol{\theta}^t \in \mathbb{SO}(3)^{23}$ is the local pose of the 23 body joints, $\boldsymbol{\beta}^t \in \mathbb{R}^{10}$ represents the body shape of the human, and $\boldsymbol{T}_g^t = (\boldsymbol{R}_g^t, \boldsymbol{t}_g^t) \in \mathbb{SE}(3)$ is the global orientation and translation of the SMPL meshes in the world coordinate. The human pose $\boldsymbol{\theta}^t$ and shape $\boldsymbol{\beta}^t$ can be obtained from human mesh recovery techniques (Goel et al., 2023), as they mainly depend on the local 2D image context. Estimating the global transformation $\boldsymbol{T}_g^t$ is more challenging as camera motion and human motion are entangled in a monocular video. We can decompose the global transformation $\boldsymbol{T}_g^t$ as $\boldsymbol{T}_g^t = \boldsymbol{P}^t \cdot \boldsymbol{T}_c^t$, where $\boldsymbol{P}^t \in \mathbb{SE}(3)$ is the camera extrinsic matrix and $\boldsymbol{T}_c^t = (\boldsymbol{R}_c^t, \boldsymbol{t}_c^t) \in \mathbb{SE}(3)$ is the transformation matrix of the SMPL meshes in the local camera coordinate. It is also possible to estimate $\boldsymbol{T}_c^t$ directly from an image (Goel et al., 2023) given the focal length of the camera and the human shape prior. Therefore, an accurate reconstruction of the global human motion depends on precise estimation of both the camera poses $\{\boldsymbol{P}^t\}_{t=1}^N$ and local SMPL parameters $\{\mathbf{\Theta}_c^t\}_{t=1}^N$ with $\mathbf{\Theta}_c^t = \{\boldsymbol{T}_c^t, \boldsymbol{\theta}^t, \boldsymbol{\beta}^t\}$.

## 2.2 DENSE SCENE RECONSTRUCTION

In dense scene reconstruction, the goal is to recover the 3D geometry of the environment given by the dense point cloud $\boldsymbol{X} \in \mathbb{R}^{N \times h \times w \times 3}$. $\boldsymbol{X}$ consists of the transformed local point map $\boldsymbol{X}^t$ of each timestep in the world coordinate:

$$\boldsymbol{X} = \bigcup_{t=1}^N \{\sigma^t \boldsymbol{P}^t \boldsymbol{X}^t | \boldsymbol{X}^t = \pi^{-1}(\boldsymbol{K}^t, \boldsymbol{Z}^t)\}, \tag{1}$$

where $\sigma^t \in \mathbb{R}$ is the scale for each frame, $\boldsymbol{K}^t \in \mathbb{R}^{3 \times 3}$ is the camera intrinsic matrix, $\boldsymbol{Z}^t \in \mathbb{R}^{h \times w}$ is the dense depth map for each frame, and $\pi^{-1}(\cdot)$ is the unprojection from 2D pixels of the frame to 3D points in camera coordinate given the camera intrinsics and depth map. A common way for dense scene reconstruction is to first predict the per-frame point maps $\boldsymbol{X}^t$ in the local camera coordinate and the point correspondences between adjacent frames $\boldsymbol{C} = \{(\boldsymbol{x}^i, \boldsymbol{x}^j, c) | \boldsymbol{x}^i \in \boldsymbol{X}^i, \boldsymbol{x}^j \in \boldsymbol{X}^j, (i, j) \in \mathcal{E}\}$, where $\mathcal{E}$ contains pairs of frames in the optimization graph, and $c$ is the confidence of the point correspondence. It then performs a global optimization of all parameters with the 3D correspondence loss and 2D reprojection loss:

$$\mathcal{L}_{3D} = \sum_{(\mathbf{x}^i, \mathbf{x}^j, c) \in \mathbf{C}} c \cdot \rho(\sigma^i \boldsymbol{P}^i \mathbf{x}^i - \sigma^j \boldsymbol{P}^j \mathbf{x}^j) \tag{2}$$

$$\mathcal{L}_{2D} = \sum_{(\mathbf{x}^i, \mathbf{x}^j, c) \in \mathbf{C}} c \cdot \left[ \begin{matrix} \rho(\pi(\boldsymbol{K}^i, \mathbf{x}^i) - \pi(\boldsymbol{K}^i, \frac{\sigma^j}{\sigma^i}(\boldsymbol{P}^i)^{-1} \boldsymbol{P}^j \mathbf{x}^j)) + \\ \rho(\pi(\boldsymbol{K}^j, \mathbf{x}^j) - \pi(\boldsymbol{K}^j, \frac{\sigma^i}{\sigma^j}(\boldsymbol{P}^j)^{-1} \boldsymbol{P}^i \mathbf{x}^i)) \end{matrix} \right] \tag{3}$$

$$\mathcal{L}_{scene} = w_{3D} \cdot \mathcal{L}_{3D} + w_{2D} \cdot \mathcal{L}_{2D}, \tag{4}$$

where $\pi(\cdot)$ projects 3D points in the camera coordinate to the pixel locations with camera intrinsics and $\rho(\cdot)$ is a robust loss to deal with potential outliers.

## 3 JOINT OPTIMIZATION OF SCENE AND HUMAN

Humans are constantly interacting with the scene. The moving camera captures such interaction and encodes rich geometry information useful to both human and scene reconstruction. Therefore, it is necessary to perform joint optimization for 4D human-scene reconstruction as it enables the camera, the human, and the scene to benefit each other continuously. Existing methods (Shin et al., 2024; Xue et al., 2024; Zhao et al., 2024) handle scene reconstruction, human motion estimation, and camera pose estimation as separate tasks or perform separate optimization. This separation can lead to inconsistent and inaccurate reconstruction results. To better exploit the synergy between the camera, the human, and the scene, we introduce JOSH, a general optimization framework that jointly optimizes all the parameters for 4D human-scene reconstruction in a single stage, allowing all tasks to mutually refine each other. An overview of JOSH is shown in Fig. 2. We first formalize the task of 4D human-scene reconstruction from monocular videos in Sec. 3.1. We then discuss the initialization

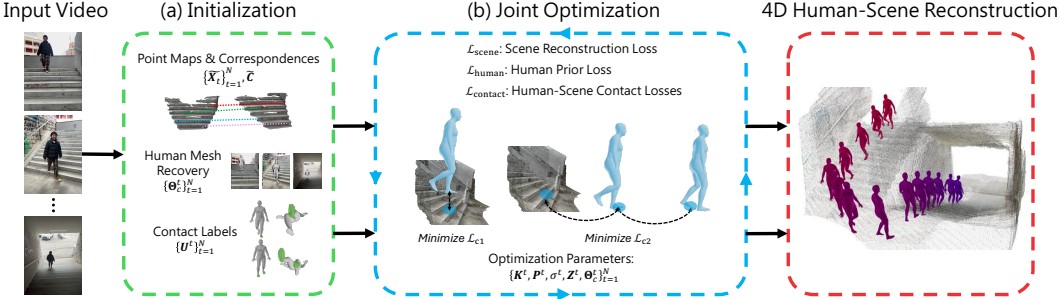

Figure 2: **Overview of JOSH**. Given an input video, JOSH first initializes the point maps, point correspondences, local human mesh recovery, and contact labels from pre-trained models. JOSH then jointly optimizes the camera poses, the dense scene point cloud, and the global human motion with the key human-scene contact losses to predict 4D human-scene reconstruction.

before optimization in Sec. 3.2. We introduce our joint optimization procedure, including the key human-scene contact losses in Sec. 3.3. We finally present JOSH3R, an end-to-end prediction model in Sec. 3.4.

## 3.1 PROBLEM DEFINITION

We aim to tackle the problem of 4D human-scene reconstruction, which reconstructs the global human motion of all humans, the surrounding scene context, and the camera parameters from in-the-wild videos. The input is a single monocular video with a sequence of images $\{\mathbf{I}^t \in \mathbb{R}^{h \times w \times 3}\}_{t=1}^N$, where $N$ is the total number of frames. The optimization parameters are $\{\mathbf{K}^t, \mathbf{P}^t, \sigma^t, \mathbf{Z}^t, \{\mathbf{\Theta}_{c,k}^t\}_{k=1}^M\}_{t=1}^N$, including the camera intrinsic and extrinsic parameters, the dense depth maps with global scales, and the local SMPL parameters for each human $k$ among all $M$ humans. The final output is the joint reconstruction of the dense scene point cloud $\mathbf{X}$ and the global human motion $\{\{\mathbf{\Theta}_{g,k}^t\}_{k=1}^M\}_{t=1}^N$. For notation simplicity, we will omit the subscript $k$ when discussing the human parameters, as they apply to all humans during joint optimization.

## 3.2 INITIALIZATION

JOSH uses off-the-shelf models to initialize parameters for optimization. We use dense scene reconstruction methods to initialize the local point maps $\{\mathbf{X}^t\}_{t=1}^N$ and point correspondences $\mathbf{C}$. As many dense scene reconstruction methods only work for static environments, humans in monocular videos may introduce additional noise within reconstructed depth maps and result in wrong correspondences, leading to degraded reconstruction performance. Hence, we first segment out the moving humans using a video segmentation model DEVA (Cheng et al., 2023), resulting in 2D masks $\mathbf{M}^t$ for each image. We then trim the point cloud and matching results based on the masks, i.e., $\tilde{\mathbf{X}}^t = \{\mathbf{x}^t \in \mathbf{X}^t | \pi(\mathbf{K}^t, \mathbf{x}^t) \in \mathbf{M}^t\}$, $\tilde{\mathbf{C}} = \{(\mathbf{x}^i, \mathbf{x}^j, c) \in \mathbf{C} | \mathbf{x}^i \in \tilde{\mathbf{X}}^i, \mathbf{x}^j \in \tilde{\mathbf{X}}^j\}$, and use only the trimmed results with points in the background for scene reconstruction. We also use human mesh recovery methods to provide an initial estimation of the local SMPL parameters $\{\mathbf{\Theta}_c^t\}_{t=1}^N$ for all humans in the video. To compute the key human-scene contact losses, we finally predict the per-vertex contact labels of the SMPL mesh for all humans $\{\mathbf{U}^t \subset \mathbf{V}_c^t\}_{t=1}^N$ from the image using a contact prediction model BSTRO (Huang et al., 2022).

JOSH is compatible with different dense scene reconstruction and human mesh recovery methods as long as they can provide the required initializations. In our experiments, we will show the benefit of joint optimization with different initializations.

## 3.3 JOINT OPTIMIZATION

Human-scene contact is the most natural form of human-scene interaction. It can provide explicit geometry constraints and physical grounding that bridge camera pose estimation, human motion estimation, and scene reconstruction with joint optimization. These contact constraints can help refine body pose in motion estimation, ensuring physically plausible interactions with the environment while improving the accuracy of the scene geometry and the camera poses. Therefore, we aim to

leverage human-scene contact by inferring two key human-scene contact losses for joint optimization given the estimated contact labels $\{\boldsymbol{U}^t\}_{t=1}^N$.

**Contact Scene Loss** Human-scene contact can help identify corresponding contact points from the human and the scene, and hence, we can ensure the physical plausibility of the contact by constraining corresponding contact points from the human body mesh and the dense scene point cloud to be close in the final reconstruction with the contact scene loss $\mathcal{L}_{c1}$. We first identify the correspondence between scene contact points $\boldsymbol{x}_s^t$ in the background scene point cloud and the predicted human contact points $\boldsymbol{x}_h^t$. For all predicted contact vertices $\boldsymbol{x}_h^t$ of the human body mesh in initialization, we require these contact vertices to be visible to avoid depth ambiguities, i.e., $\pi(\boldsymbol{K}^t, \boldsymbol{x}_h^t) \in (1 - \boldsymbol{M}^t)$. Next, we search for the corresponding contact point in the scene background with the closest distance to the projected contact vertex: $\boldsymbol{x}_s^t = \arg\min_{\boldsymbol{x}^t \in \tilde{\boldsymbol{X}}^t} |\pi(\boldsymbol{K}^t, \boldsymbol{x}^t) - \pi(\boldsymbol{K}^t, \boldsymbol{x}_h^t)|_2$. We finally do a filtering, requiring that the depth predictions of the scene contact point and the human contact point are close in a monocular depth prior $\hat{\boldsymbol{Z}}^t$ (Bhat et al., 2023): $|\hat{\boldsymbol{Z}}^t(\pi(\boldsymbol{K}^t, \boldsymbol{x}_s^t)) - \hat{\boldsymbol{Z}}^t(\pi(\boldsymbol{K}^t, \boldsymbol{x}_h^t))| < \epsilon$. We empirically find that such a search heuristic generally leads to correct contact correspondences, especially for foot and hand contacts. Then we can obtain the contact correspondences $\boldsymbol{D} = \{(\boldsymbol{x}_h^t, \boldsymbol{x}_s^t) | \boldsymbol{x}_h^t \in \boldsymbol{U}^t, \boldsymbol{x}_s^t \in \tilde{\boldsymbol{X}}^t\}$, and the contact scene loss $\mathcal{L}_{c1}$ can be computed as follows:

$$\mathcal{L}_{c1} = \sum_{(\boldsymbol{x}_h^t, \boldsymbol{x}_s^t) \in \boldsymbol{D}} \rho(\boldsymbol{x}_h^t - \sigma_t \boldsymbol{x}_s^t). \tag{5}$$

Note that as the human contact point $\boldsymbol{x}_h^t$ is updated during joint optimization, we search for and update the corresponding scene contact point $\boldsymbol{x}_s^t$ *in each iteration*.

**Contact Static Loss** Human-scene contact can also help identify body parts that are relatively static to the scene when a contact is maintained across frames. This motivates us to design the contact static loss $\mathcal{L}_{c2}$ that encourages such contact remains static to reduce sliding motion. We can check if the same vertex remains in contact in adjacent frames and denote the results as $\boldsymbol{E} = \{(\boldsymbol{x}_h^i, \boldsymbol{x}_h^j) | \boldsymbol{x}_h^i \in \boldsymbol{U}^i, \boldsymbol{x}_h^j \in \boldsymbol{U}^j\}$. $\mathcal{L}_{c2}$ ensures both the human contact points and their corresponding scene contact points remain static as follows:

$$\mathcal{L}_{c2} = \sum_{(\boldsymbol{x}_h^i, \boldsymbol{x}_h^j) \in \boldsymbol{E}} (\rho(\boldsymbol{P}^i \boldsymbol{x}_h^i - \sigma_j \boldsymbol{P}^j \boldsymbol{x}_s^j) + \rho(\boldsymbol{P}^j \boldsymbol{x}_h^j - \sigma_i \boldsymbol{P}^i \boldsymbol{x}_s^i)). \tag{6}$$

**Final Loss** The final loss for joint optimization includes the scene reconstruction loss, the human prior loss, and the key human-scene contact loss. The scene reconstruction loss $\mathcal{L}_{scene}$ is computed from point correspondences $\tilde{\boldsymbol{C}}$ in the static background using Eqn. 4. The human prior loss $\mathcal{L}_{human}$ consists of a temporal smoothness loss to ensure smoothness of the global human motion, a SMPL prior loss to constrain the estimated SMPL local parameters close to their initial values, and a 2D key-point reprojection loss to regularize the 2D projection of the SMPL joints:

$$\mathcal{L}_{human} = \sum_{t=1}^N (w_{smooth} |\boldsymbol{\Theta}_g^t - \boldsymbol{\Theta}_g^{t+1}| + w_{smpl} |\boldsymbol{\Theta}_c^t - \hat{\boldsymbol{\Theta}}_c^t|_2 + w_{2dkp} |\boldsymbol{J}_{2D}^t - \hat{\boldsymbol{J}}_{2D}^t|), \tag{7}$$

where $\hat{\boldsymbol{\Theta}}_c^t$ is the initial local SMPL parameters from human mesh recovery, $\boldsymbol{J}_{2D}^t$ is the projected SMPL joints, and $\hat{\boldsymbol{J}}_{2D}^t$ is the estimated 2D keypoints using ViTPose (Xu et al., 2022). Finally, we use the key human-scene contact losses $\mathcal{L}_{contact}$ to guide both human and scene reconstruction with $\mathcal{L}_{contact} = w_{c1} \mathcal{L}_{c1} + w_{c2} \mathcal{L}_{c2}$. We sum up all aforementioned losses to obtain our final loss:

$$\mathcal{L} = \mathcal{L}_{scene} + \mathcal{L}_{human} + \mathcal{L}_{contact}. \tag{8}$$

We use this loss to optimize *all the parameters* $\{\boldsymbol{K}^t, \boldsymbol{P}^t, \sigma^t, \boldsymbol{Z}^t, \boldsymbol{\Theta}_c^t = (\boldsymbol{T}_c^t, \boldsymbol{\theta}^t, \boldsymbol{\beta}^t)\}_{t=1}^N$ *in a single stage* with a gradient based optimizer. Note that the final result is *on a metric scale* as the human prior loss $\mathcal{L}_{human}$ encodes information about the global scale in metric units.

**Optimizing Focal Length** As many in-the-wild videos don't provide camera intrinsics information $\boldsymbol{K}^t$, prior works (Ye et al., 2023; Zhao et al., 2024; Wang et al., 2024b) approximate a fixed focal length such as the diagonal pixel length. However, as the root depth of the local human mesh $t_z$ predicted by existing human mesh recovery models (Goel et al., 2023; Cai et al., 2023) is proportional to focal length $f$, a wrong initial focal length estimation would also lead to an irrecoverable error

in the human motion. On the contrary, JOSH supports optimizing the focal length of the camera $f$ while updating the local root depth thanks to joint optimization. To be specific, we add $f$ to the optimization parameters and set the local translation of the SMPL mesh $\boldsymbol{t}_c^t = [t_x, t_y, t_z']$ in each optimization iteration, where $t_z' = \frac{f}{f_{\text{init}}} t_z$ and $f_{\text{init}}$ is the initial focal length. This formulation ensures that the depth adjustment remains consistent with the focal length update, leading to more accurate 4D human-scene reconstruction. We show the benefit of jointly optimizing focal length in Sec. 4.5.

## 3.4 END-TO-END PREDICTION WITH JOSH3R

Web videos provide a diverse and large-scale data source for real-world global human motion, but their unstructured nature makes it challenging to get reliable ground-truth annotations. On the other hand, existing real-world global human motion datasets (Dai et al., 2023; Kaufmann et al., 2023; Huang et al., 2022) are small in scale and lack diverse scene backgrounds, real-world motion, and camera movements, making it challenging to train an efficient end-to-end model for recovering global human motion in the world coordinate directly from monocular videos. Due to the strong performance of JOSH in in-the-wild settings as shown in Fig. 3 (b), it can generalize to noisy web videos and enable scalable training of end-to-end models from extensive web videos. To this end, we use JOSH to label global human motion from about 20 hours of web videos and train an end-to-end model JOSH3R. Inspired by the strong performance of MASt3R Leroy et al. (2024) in geometric scene understanding, we introduce a new human trajectory head to output the relative local human transformation $\Delta \boldsymbol{T}_c^i$ between adjacent frames, which is in parallel with the original output heads for local scene reconstruction. Assuming the origin of the world coordinate aligns with the camera coordinate of the first frame, the global human transformation and the global camera pose can then be computed iteratively without optimization as:

$$\boldsymbol{T}_g^t = \prod_{i=1}^{t-1} \Delta \boldsymbol{T}_c^i \cdot \boldsymbol{T}_c^1, \boldsymbol{P}^t = \boldsymbol{T}_g^t \cdot (\boldsymbol{T}_c^t)^{-1}, \tag{9}$$

where $\boldsymbol{T}_c^1$ is the local transformation of the SMPL human mesh in the first frame. We leave the implementation details of labeling web videos and JOSH3R in the supplementary material.

## 4 EXPERIMENTS

We discuss our experiment setups in Sec. 4.1. We show the effectiveness of JOSH on 4D human-scene reconstruction in Sec. 4.2. Then, we show the benefit of JOSH on global human motion estimation and dense scene reconstruction respectively in Sec. 4.3 and Sec. 4.4. Ablation studies on JOSH can be found in Sec. 4.5. We further show the potential of JOSH to enable scalable training of JOSH3R from web videos in Sec. 4.6. Implementation details about JOSH can be found in Appendix B.

## 4.1 EXPERIMENT SETUPS

**Datasets** We use the SLOPER4D dataset (Dai et al., 2023) as the main dataset for ablation and comparison, as it provides ground truth labels for the global human motion, the scene mesh, and the camera parameters annotated from LiDAR point clouds, and we use the 6 publicly released sequences for evaluation. We also provide results on the EMDB (Kaufmann et al., 2023) and RICH (Huang et al., 2022) datasets. The EMDB dataset only provides labels of the global human motion and camera poses without the scene labels, and we use the EMDB-2 subset with 25 sequences for evaluation following previous works (Shin et al., 2024; Wang et al., 2024b). The RICH dataset only provides global human motion labels and pre-captured scene meshes from a laser scanner without ground-truth camera parameters, and we use the subset of 40 sequences with a moving camera.

**Metrics** We evaluate on the following aspects of 4D human-scene reconstruction: global human motion estimation accuracy, dense scene reconstruction quality, and physics plausibility. As different datasets contain different labels, we only evaluate the available metrics for each dataset. For **global human motion estimation**, we follow prior works (Ye et al., 2023; Shin et al., 2024; Wang et al., 2024b) to measure errors in the world coordinate. We split sequences into segments of 100 frames and align the predicted motion with the ground truth using either the first two frames or the entire

Table 1: **Effectiveness of JOSH on 4D human-scene reconstruction**. We compare different variants of JOSH with the baseline SynCHMR* (Zhao et al., 2024) on the SLOPER4D (Dai et al., 2023) dataset.

| Methods | Initialization | | Human Metrics | | | Scene Metrics | | | | Physics Metrics | | |
|---|---|---|---|---|---|---|---|---|---|---|---|---|
| | Human | Scene | WA-MPJPE ↓ | W-MPJPE ↓ | RTE% ↓ | ATE ↓ | Abs Rel ↓ | δ<1.25 ↑ | CD ↓ | Jitter ↓ | FS ↓ | FFR% ↓ |
| SynCHMR* | HMR2.0 | DROID-SLAM | 233.2 | 1125.4 | 4.5 | 17.47 | 0.25 | 0.55 | 17.76 | 123.9 | 67.4 | 9.0 |
| JOSH₁ | HMR2.0 | DROID-SLAM | 206.3 | 1094.2 | 4.4 | 17.18 | 0.25 | 0.57 | 16.84 | 7.6 | 56.9 | 3.3 |
| JOSH₂ | WHAM | MonST3R | 210.4 | 994.3 | 3.0 | 14.53 | 0.22 | 0.75 | 9.09 | 7.8 | 45.3 | **2.1** |
| JOSH₃ | VIMO | MASt3R | **120.0** | **438.3** | **1.7** | **3.21** | **0.16** | **0.87** | **5.31** | **7.1** | 28.2 | 2.9 |

Figure 3: **Qualitative 4D human-scene reconstruction results with the JOSH₃ variant.** a): Qualitative results on the RICH (Huang et al., 2022) dataset. We compare the ground truth reconstruction with JOSH with joint optimization and the SynCHMR* (Zhao et al., 2024) baseline without joint optimization. JOSH has better reconstruction quality and coherency between global human motion and dense scene reconstruction. b). Qualitative results of web videos. JOSH can reconstruct the motion of multiple people and their surrounding environment in the wild. More video qualitative results are included in the supplementary video.

segment and compute the mean-per-joint error in millimeters to obtain W-MPJPE or WA-MPJPE for evaluation. We also evaluate the relative Root Translation Error (RTE%) of the whole trajectory. For **dense scene reconstruction**, as JOSH can reconstruct the scene on a metric scale, we do not perform global scale alignment. Following previous literature (Wang et al., 2024a; Zhang et al., 2024; Zhao et al., 2024), we first evaluate the video depth estimation using absolute relative error (AbsRel) and the fraction of inlier points (δ<1.25) as well as the camera poses using absolute trajectory error (ATE) in meters as indirect metrics. We then evaluate the dense reconstruction with chamfer distance (CD) in meters. For **physics plausibility**, we evaluate jittering (Jitter, $10m/s^3$) and foot sliding during contact (FS, mm) for motion plausibility. Jittering measures the rate of change of 3D joint acceleration with respect to time. Foot sliding is the average foot displacement between adjacent frames where the ground truth foot displacement is less then a threshold (10mm). We also evaluate foot floating rate (FFR%) for human-scene interaction plausibility. The foot floating rate computes the minimum distance between the human foot and the scene point cloud and reports the ratio of the distance greater than a given threshold (20cm).

**Initialization Methods** To show the benefit of JOSH as a general optimization framework, we build different variants of JOSH with various state-of-the-art human mesh recovery and dense scene reconstruction methods to provide initializations as shown in Tab. 1 (column 'Initialization'). We experiment with HMR2.0 (Goel et al., 2023), WHAM (Shin et al., 2024), and VIMO (Wang et al., 2024b) to provide local human mesh initializations. For scene initialization, we experiment with DROID-SLAM (Teed & Deng, 2021), MonST3R (Zhang et al., 2024), and MASt3R (Leroy et al., 2024). We combine these initializations to get three variants as JOSH₁, JOSH₂, and JOSH₃.

## 4.2 4D HUMAN-SCENE RECONSTRUCTION

To show the effectiveness of JOSH on 4D human-scene reconstruction, we compare it with SynCHMR (Zhao et al., 2024) as our main baseline. SynCHMR performs sequential optimization of scene and human instead of joint optimization. It first calibrates the scale of the scene reconstruction and camera poses using HMR2.0 (Goel et al., 2023) and ZoeDepth (Bhat et al., 2023). It then transforms the local SMPL meshes to the world coordinates while fixing the scene reconstruc-

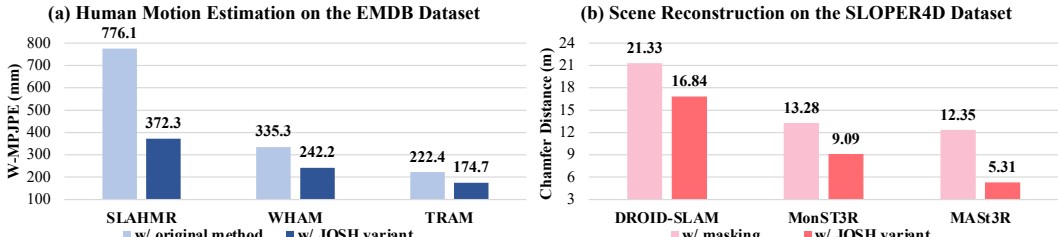

Figure 4: **Quantitative comparison results on a): global human motion estimation and b): dense scene reconstruction (↓).** For each method, we compare with the JOSH variant that uses the same local human initialization / scene initialization to highlight the effectiveness of joint optimization.

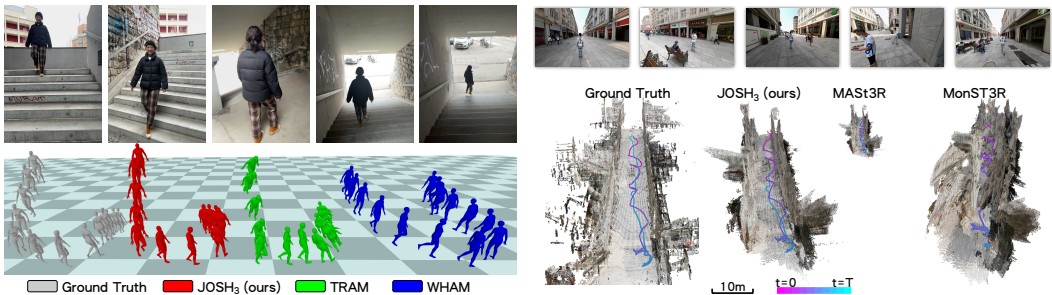

Figure 5: **Qualitative comparisons for global human motion estimation on EMDB.** JOSH has the best accuracy compared to the other baselines, while TRAM Wang et al. (2024b) has the wrong global body orientation, and WHAM Shin et al. (2024) has large errors in global translation.

Figure 6: **Qualitative comparisons for dense scene reconstruction on SLOPER4D.** We visualize the reconstructed dense scene point cloud and the camera trajectory. JOSH has the best accuracy compared to the other baselines, while MASt3R (Leroy et al., 2024) has the wrong scale, and MonST3R (Zhang et al., 2024) fails to produce a consistent reconstruction.

tion. As their implementation is not publicly available, we re-implement their method and call it SynCHMR⋆. As shown in Tab. 1, with the same initialization method, $JOSH_1$ performs better than SynCHMR⋆ in all metrics, especially in the physics plausibility, which reduces jittering from 123.9 to 7.6, foot sliding from 67.4 to 56.9, and foot floating rate from 9.0% to 3.3%. In addition, JOSH is a more general framework that can benefit from more recent works in local human mesh recovery and dense scene reconstruction. For example, when using VIMO and MASt3R as initialization, the variant $JOSH_3$ reduces error by 46.6% in WA-MPJPE, and reduces error by 70.1% in chamfer distance compared to SynCHMR⋆, while SynCHMR⋆ can only work with DROID-SLAM (Teed & Deng, 2021). As shown in Fig. 3 (a), as SynCHMR⋆ does not directly optimize the detailed contact between scene and human, the reconstruction result lacks coherency and physics plausibility with significant foot penetration in the scene, which further highlights the value of joint optimization. As JOSH is a more general method, we believe its performance can be further improved with better initializations in future models.

## 4.3 GLOBAL HUMAN MOTION ESTIMATION

We compare state-of-the-art global human motion estimation methods SLAHMR (Ye et al., 2023), WHAM (Shin et al., 2024), and TRAM (Wang et al., 2024b) with the corresponding JOSH variant that has the same local human initialization. The results are shown in Fig 4 (a). It can be observed that JOSH always leads to better performance than its counterpart method. For example, $JOSH_1$ performs better than SLAHMR (Ye et al., 2023) when both use HMR2.0 as initialization, achieving a W-MPJPE of 372.3 on the EMDB dataset compared to 776.1 with SLAHMR. In addition, when using VIMO Wang et al. (2024b) as initialization, $JOSH_3$ performs better than its counterpart TRAM and sets up a new state-of-the-art on the EMDB dataset with a 174.7 W-MPJPE. These results demonstrate the general benefit of jointly optimizing the human and the scene in obtaining an accurate global human motion estimation. Some qualitative comparisons are shown in Fig. 5. A complete comparison table on all datasets can be found in Appendix. C.

Table 2: **Ablation experiments of JOSH on SLOPER4D**. Row 1: Disabling the contact scene loss $\mathcal{L}_{c1}$. Row 2: Not optimizing the local human motion $\mathbf{\Theta}_c$. Row 3: Disabling the contact static loss $\mathcal{L}_{c2}$. Row 4: The full optimization with the JOSH$_3$ variant. Row 5 and Row 6: Optimizing without the given ground truth camera intrinsics, while Row 6 also optimizes the intrinsics.

| | Human | | Scene | | Physics | |
|---|---|---|---|---|---|---|
| Variants | W-MPJPE↓ | RTE%↓ | Abs Rel↓ | ATE↓ | FS↓ | FFR%↓ |
| -$\mathcal{L}_{c1}$ | 1361.4 | 4.7 | 0.49 | 22.47 | 47.3 | 92.9 |
| -opt $\mathbf{\Theta}_c$ | 486.4 | **1.8** | **0.17** | 3.28 | 35.6 | 3.2 |
| -$\mathcal{L}_{c2}$ | 448.3 | 1.9 | 0.18 | 3.26 | 68.2 | 3.2 |
| JOSH$_3$ | **438.3** | **1.8** | **0.17** | **3.21** | **28.2** | **2.9** |
| Fixed Intrin. | 1220.7 | 5.5 | 0.60 | 19.89 | 71.3 | 15.2 |
| Opt. Intrin. | **1053.4** | **4.6** | **0.47** | **16.91** | **68.3** | **6.8** |

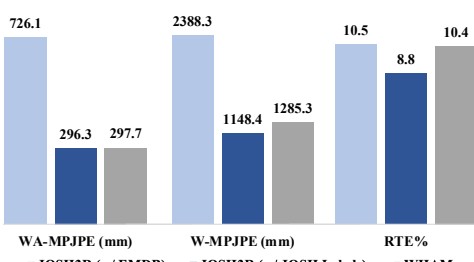

Figure 7: **Scalable training results with JOSH labels (↓).** We compare to JOSH3R trained on the EMDB (Kaufmann et al., 2023) dataset and WHAM (Shin et al., 2024) on the SLOPER4D (Dai et al., 2023) dataset.

## 4.4 Dense Scene Reconstruction

Similar to global human motion estimation, we compare state-of-the-art scene reconstruction methods DROID-SLAM (Teed & Deng, 2021), MonST3R (Zhang et al., 2024), and MASt3R (Leroy et al., 2024) with the corresponding JOSH variant that has the same scene initialization in Fig. 4 (b). For a fair comparison, we also perform human masking for MASt3R and DROID-SLAM, which cannot work with dynamic objects, so that the only varying factor is joint optimization. The results show the joint optimization of JOSH can benefit different dense reconstruction methods as well. For example, JOSH$_3$ achieves 57.0% less chamfer distance on the SLOPER4D dataset compared to the original MASt3R, which only optimizes the scene. Also, we can see that the variant JOSH$_3$ achieves the best performance on both global human motion estimation and dense scene reconstruction. This further suggests a strong correlation between the two tasks and the fact that performing joint optimization is necessary. Some qualitative comparisons are shown in Fig. 6. A complete comparison table on all datasets can be found in Appendix. D.

## 4.5 Ablation Studies

We conduct ablation studies in Tab. 2 to analyze the benefits of each optimization component of JOSH using the JOSH$_3$ variant. From the comparison between Row 1 and Row 4, we show that enforcing consistency between the scene contact point and human contact point with the contact scene loss $\mathcal{L}_{c1}$ can significantly improve the performance in all metrics. For example, RTE improves from 4.7% to 1.8%, ATE improves from 22.47 to 3.21, and FFR has been reduced drastically from 92.9% to 2.9%, as the contact scene loss $\mathcal{L}_{c1}$ can guide toward the metric scale of the camera poses and the depth map, as well as improve the global human motion accuracy with better camera poses, thus boosting human-scene interaction plausibility. From the comparison between Row 2 and Row 3, we show that jointly optimizing the human parameters $\mathbf{\Theta}_c$ also leads to a better performance compared to only optimizing the scene and the camera pose, improving W-MPJPE from 486.4 to 438.3, which further underscores the significance of joint optimization. From the comparison between Row 3 and Row 4, we show that the contact static loss $\mathcal{L}_{c2}$ is useful in refining reconstruction quality by enforcing contact correspondences to remain static across time steps, which reduces foot sliding from 68.2 to 28.2. From the comparison between Row 5 and Row 6, we use the estimated camera intrinsics instead of the provided ground truth intrinsics. We can see that jointly optimizing the camera focal length $f$ can lead to better reconstruction performance in all metrics, improving W-MPJPE from 1220.7 to 1053.5, ATE from 19.89 to 16.91, and FFR from 15.2% to 6.8%. These results show the importance of jointly optimizing camera intrinsics with JOSH for in-the-wild settings, where the ground truth is often unavailable.

## 4.6 Scalable Training of JOSH3R on Web Videos

As shown in Fig. 7, we experimented with training JOSH3R using ground-truth labels from EMDB (Kaufmann et al., 2023), compared to scalable training on web videos. The results showed that JOSH3R trained with web videos annotated by JOSH surpasses the performance of the one trained on the ground truth dataset by a large margin, improving WA-MPJPE by a significant 59.2%. It also achieves comparable performance to the state-of-the-art global human motion estimation

Table 3: **Performance analysis of JOSH and JOSH3R**. We compare the efficiency and accuracy of the JOSH$_3$ variant and JOSH3R trained with JOSH labels. "FPS" indicates the amortized frames per second to run inference with all modules of the method. All methods are tested on an Nvidia RTX 4090 GPU. We evaluate on the EMDB (Kaufmann et al., 2023), SLOPER4D (Dai et al., 2023), and RICH (Huang et al., 2022) datasets.

| Methods | FPS↓ | EMDB | | | SLOPER4D | | | RICH | | |
|---|---|---|---|---|---|---|---|---|---|---|
| | | WA-MPJPE↓ | W-MPJPE↓ | RTE%↓ | WA-MPJPE↓ | W-MPJPE↓ | RTE%↓ | WA-MPJPE↓ | W-MPJPE↓ | RTE%↓ |
| SLAHMR | 0.1 | 326.9 | 776.1 | 10.2 | 706.9 | 3401.0 | 12.6 | 132.2 | 237.1 | 6.4 |
| WHAM | 4.4 | 131.1 | 335.3 | 4.1 | 297.7 | 1272.3 | 10.5 | 108.4 | 196.1 | 4.5 |
| TRAM | 1.1 | 76.4 | 222.4 | 1.4 | 215.1 | 1285.3 | 3.0 | 127.8 | 238.0 | 6.0 |
| JOSH$_3$ | 0.8 | **68.9** | **174.7** | **1.3** | **120.0** | **438.3** | **1.8** | **102.6** | **184.3** | **3.4** |
| JOSH3R | **15.4** | 220.0 | 661.7 | 13.1 | 296.3 | 1148.4 | 10.4 | 190.4 | 334.9 | 6.3 |

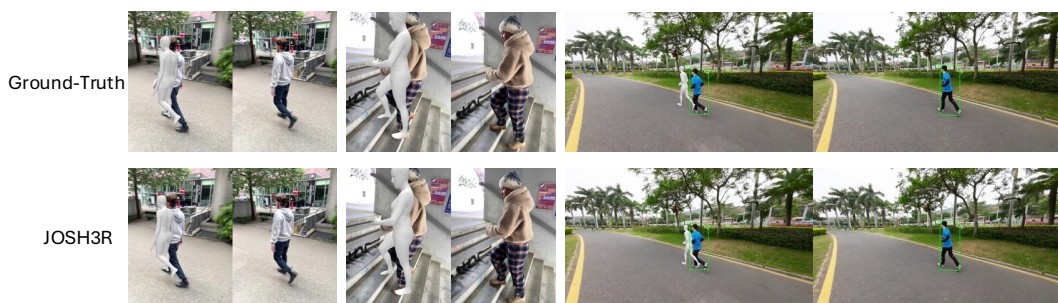

Figure 8: **Qualitative results of JOSH3R.** We visualize the human pose of the right (future) frame in the left (current) frame's coordinate.

method WHAM (Shin et al., 2024) with a more accurate human trajectory in terms of RTE (8.8% vs. 10.4%). Considering that WHAM uses extensive ground-truth human motion data (Mahmood et al., 2019) for training, JOSH3R's high performance reflects JOSH's ability to generalize well to in-the-wild settings and label diverse and realistic global human motion from web videos without manual effort. We believe the performance of JOSH3R can be further improved by scaling up web data annotated by JOSH, and we will leave it to future work.

To further analyze the accuracy and efficiency of JOSH and JOSH3R, we conduct experiments in Tab. 3. Since JOSH3R takes a data-driven approach, it performs best on the SLOPER4D dataset, which is most similar to pedestrian street walking videos from JOSH labels. We can observe that while JOSH achieves the best accuracy, JOSH3R is significantly faster than JOSH, with the best efficiency of 15.4 FPS vs. 0.8 FPS, allowing real-time inference. Although JOSH3R's accuracy is currently only on par with traditional approaches, these results highlight a promising path toward training efficient, end-to-end global human motion estimation models using pseudo-labels and strong visual geometry backbones. Some qualitative results of JOSH3R are shown in Fig. 8.

## 5 CONCLUSION

We present JOSH, a general optimization framework for 4D human-scene reconstruction in the wild. JOSH leverages human-scene contact to jointly optimize the camera pose, the global multi-human motion, and the scene geometry. The results show that the joint optimization of JOSH can not only produce coherent human-scene interaction data but also achieve better performance in both global human motion estimation and dense scene reconstruction. We believe JOSH and JOSH3R could enable scalable training of 4D human-scene reconstruction on web videos with real-world human motion and diverse environments.

**Limitations.** JOSH uses results from off-the-shelf models as initialization. Therefore, JOSH will benefit from future models with better performance, while it could suffer from bad initializations even after joint optimization. In addition, JOSH works best when both the human and scene contact points are visible in the 2D image, which could fail in some more challenging settings when the contact correspondences are hard to obtain or there is no contact between the human and the scene.

## ETHICS STATEMENT

Our work focuses on reconstructing human motion and surrounding environments from monocular in-the-wild videos. We are mindful of ethical considerations regarding the use of human data. All datasets employed in this study (SLOPER4D (Dai et al., 2023), EMDB (Kaufmann et al., 2023), and RICH (Huang et al., 2022)) are publicly released for research purposes with appropriate usage licenses, and we adhere strictly to their terms of use. When leveraging additional web videos for large-scale training, we only utilize publicly available, non-identifiable footage and avoid data containing sensitive or private information. The reconstructed human motion data are treated strictly as research outputs for advancing computer vision and graphics techniques, and we do not use or intend them for surveillance, biometric identification, or any application that could compromise individual privacy or safety. Finally, our models are designed to study human–scene interaction in general, rather than focusing on specific demographic or personal attributes, thus limiting risks of discriminatory outcomes.

## REPRODUCIBILITY STATEMENT

We have made careful efforts to ensure the reproducibility of our results. All datasets used in this paper are publicly available benchmarks (SLOPER4D (Dai et al., 2023), EMDB (Kaufmann et al., 2023), RICH (Huang et al., 2022)) following standard evaluation protocols from prior works on global human motion estimation (Shin et al., 2024) and dense scene reconstruction (Zhang et al., 2024). We detail our optimization objectives, initialization choices, and ablation settings in Sec. 3 and Sec. 4. Hyperparameters, loss weights, and implementation details are specified in Appendix B. To foster reproducibility, we will release the source code of JOSH and scripts for dataset preprocessing and evaluation upon publication. Moreover, we provide a comprehensive comparison across different initialization methods on various datasets using different metrics in Appendices C and D to demonstrate the robustness of our framework under varying experimental conditions. Together, these measures should allow other researchers to faithfully reproduce and extend our results.

## ACKNOWLEDGMENTS

The project was supported by NSF Grants CNS-2235012, CCF-2344955, and by the ONR grant N000142512166.

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

# Appendix

We discuss related works in Sec. A. We provide implementation details of JOSH in Sec. B. More results on global human motion estimation are shown in Sec. C. More results on dense scene reconstruction are shown in Sec. D. More comparison on the EMDB (Kaufmann et al., 2023) dataset is presented in Sec. E. Sec. F shows an analysis of JOSH's robustness to noisy initializations. We then discuss the procedure to label web videos using JOSH for training JOSH3R in Sec. G. The architecture of JOSH3R and its implementation details are discussed in Sec. H. Additional qualitative results, contact label visualizations, and failure cases can be found in the supplementary video.

**We do not use Large Language Models (LLMs) in paper writing.**

## A    RELATED WORK

**Monocular Global Human Motion Estimation.**    Estimating global human motion in the world coordinates from a single monocular video is a challenging task as the camera motion and the human motion are entangled in the video, and we need to recover both motions in metric units. GLAMR and HuMoR(Rempe et al., 2021; Yuan et al., 2022) propose to leverage the motion prior to human poses and predict the ego movements between frames, leading to noisy and unreliable predictions, especially for the rare poses. An alternative solution is to estimate the camera motion with SLAM (Teed & Deng, 2021) and then transform the camera-coordinate motion to the global coordinate with the estimated camera poses. The key challenge here is to resolve the scale ambiguity of the poses: TRAM (Wang et al., 2024b) proposes to approximate scale from monocular depth estimation, WHAC (Yin et al., 2024) gauges the scale from the human motion, and OfCam (Yang et al., 2024) estimates the scale from human mesh depth. SLAHMR (Ye et al., 2023) proposes an optimization pipeline that jointly optimizes the scale and human meshes using both the human motion prior and SLAM camera poses. WHAM (Shin et al., 2024) performs such optimization implicitly by taking the SLAM camera angular velocity as input and learning the human motion prior from 3D keypoint sequences of AMASS (Mahmood et al., 2019). In contrast, COIN (Li et al., 2024) learns the motion prior from a diffusion model. Though many methods (Shin et al., 2024; Wang et al., 2024b; Ye et al., 2023) include a SLAM module in their pipeline, they treat SLAM as a stand-alone module and do not optimize the scene, the human motion, and the camera motion in a single stage. Unlike existing approaches, our method jointly optimizes the dense scene background, the foreground human motion, and the camera motion for 4D human-scene reconstruction. JOSH also supports jointly optimizing the motion of multiple people simultaneously, while many state-of-the-art approaches (Wang et al., 2024b; Shin et al., 2024; Shen et al., 2024) focus only on reconstructing the motion of one person.

**Human-Scene Interaction and Reconstruction.** Existing human-scene interaction datasets (Huang et al., 2022; Dai et al., 2022; Yan et al., 2023; 2024; Dai et al., 2023; Jiang et al., 2024) set up additional sensors like multi-view RGBD cameras, laser scanners, and IMUs or use multiple shots of the same scene (Pavlakos et al., 2022) to first get ground-truth 3D labels of the environment. The human-scene interaction can then be obtained by fitting SMPL human meshes with the ground-truth scene reconstruction and camera poses (Hassan et al., 2019; Zhang et al., 2021). While these approaches can provide accurate results, they are not scalable to casual web videos where both the dense 3D scene and human motion must be estimated from a single camera. Only a few works (Luvizon et al., 2023; Zhao et al., 2024; Liu et al., 2021; Xue et al., 2024) have been proposed to tackle the challenging task of monocular 4D human-scene reconstruction, but they perform sequential optimization by reconstructing the camera pose, the scene, and the global human motion separately. Such separate optimization does not fully exploit the synergy between the scene and the human compared to our proposed joint optimization. Müller et al. (2024) performs joint optimization but only reconstructs static scenes with humans from multi-view cameras. Other recent works address the challenging task of 4D dynamic scene reconstruction with moving objects, where Zhang et al. (2024); Wang et al. (2025a) proposes to learn the dynamic scenes with an end-to-end model, and Wu et al. (2024) reconstructs the scene with dynamic 4D Gaussians Splatting. These methods aim to reconstruct general dynamic scenes without treating humans separately, while our method focuses on reconstructing scenes together with the global human motion as fine-grained SMPL meshes. In addition, most prior scene reconstruction methods can only recover the scene geometry up to an unknown scale factor, while JOSH can reconstruct the scene on a metric scale.

Table 4: **Effectiveness of JOSH on global human motion estimation**. We evaluate on the EMDB (Kaufmann et al., 2023), SLOPER4D (Dai et al., 2023), and RICH (Huang et al., 2022) datasets and compare different variants of JOSH with methods using the same local human mesh recovery initializations.

| Methods | EMDB | | | SLOPER4D | | | RICH | | |
|---|---|---|---|---|---|---|---|---|---|
| | WA-MPJPE↓ | W-MPJPE↓ | RTE%↓ | WA-MPJPE↓ | W-MPJPE↓ | RTE%↓ | WA-MPJPE↓ | W-MPJPE↓ | RTE%↓ |
| HMR2.0 + SLAHMR | 326.9 | 776.1 | 10.2 | 706.9 | 3401.0 | 12.6 | 132.2 | 237.1 | 6.4 |
| HMR2.0 + JOSH$_1$ | **130.8** | **372.3** | **6.3** | **206.3** | **1094.2** | **4.4** | **101.3** | **183.6** | **2.7** |
| WHAM | 131.1 | 335.3 | 4.1 | 297.7 | 1272.3 | 10.5 | 108.4 | 196.1 | 4.5 |
| WHAM + JOSH$_2$ | **103.0** | **242.1** | **2.4** | **210.4** | **994.3** | **3.0** | **92.1** | **158.5** | **3.7** |
| VIMO + TRAM | 76.4 | 222.4 | 1.4 | 215.1 | 1285.3 | 3.0 | 127.8 | 238.0 | 6.0 |
| VIMO + JOSH$_3$ | **68.9** | **174.7** | **1.3** | **120.0** | **438.3** | **1.8** | **102.6** | **184.3** | **3.4** |

## B  IMPLEMENTATION DETAILS

The loss weights are set to be $w_{2D} = 1, w_{3D} = 1, w_{c1} = 1, w_{c2} = 10, w_{smpl} = 10, w_{smooth} = 0.1, w_{2dkp} = 0.01$ in all experiments by grid search. The depth filtering threshold $\epsilon$ is set to 0.2m. We use Adam (Kingma, 2014) as our gradient-based optimizer implemented in PyTorch (Paszke et al., 2019), and the learning rate is set to 0.05, with the number of iterations set to 1000. As it is slow and computationally expensive to optimize the dense scene point cloud of the entire long sequence, we split each sequence into segments of 100 frames to perform local reconstruction and concatenate the results afterward. To further improve the processing speed, we sample the keyframe every 0.2s and interpolate the camera poses in between.

To get scene initializations with different variants of JOSH, we use depth and optical flow maps from DROID-SLAM (Teed & Deng, 2021) to get initial point maps and correspondences. We can directly obtain point maps and correspondences from MonST3R (Zhang et al., 2024) and MASt3R (Leroy et al., 2024). For comparison in dense scene reconstruction, as DROID-SLAM does not directly provide dense scene reconstruction in metric scale, we combine its camera poses with ZoeDepth (Bhat et al., 2023) and scale the pose accordingly similar to TRAM (Wang et al., 2024b) to get the dense reconstruction results. MASt3R can provide metric-scale reconstruction without scaling. For MonST3R, we scale the reconstruction results according to the local human depth to get metric-scale results. For scene reconstruction evaluation on SLOPER4D, we find all methods will fail on the challenging "seq009" sequence and lead to a significantly larger error than the other sequences. We hence exclude it from the scene reconstruction evaluation so the results are not dominated by a single outlier case.

## C  MORE RESULTS ON GLOBAL HUMAN MOTION ESTIMATION

We provide additional results on global human motion estimation on all datasets in Tab. 4. For a fair comparison, we compare each method to the JOSH variant with the same local human initialization. The results show that JOSH always leads to better performance than its counterpart method on all metrics. For example, JOSH$_1$ achieves a 206.3 WA-MPJPE on the SLOPER4D dataset, outperforming SLAHMR (Ye et al., 2023), which both uses HMR2.0 (Goel et al., 2023) as initialization. JOSH$_3$ improves RTE from 6.0% to 3.4% on the RICH dataset, outperforming its counterpart TRAM (Wang et al., 2024b), which both uses VIMO (Wang et al., 2024b) as local human initialization. These results further demonstrate the general benefit of jointly optimizing the human and the scene in obtaining an accurate global human motion estimation.

## D  MORE RESULTS ON DENSE SCENE RECONSTRUCTION

We provide additional results on global human motion estimation on all datasets in Tab. 4. For a fair comparison, we compare each method to the JOSH variant with the same scene initialization. The results show that the joint optimization of JOSH can benefit different dense reconstruction methods as well, especially on the most important Chamfer distance metric. For example, JOSH$_1$ with DROID-SLAM (Teed & Deng, 2021) as scene initialization achieves a 1.57 Chamfer distance on the RICH dataset compared to the baseline of 1.81 Chamfer distance without human optimization. JOSH$_2$ using MonST3R (Zhang et al., 2024) as scene initialization also produces more accurate camera poses than the baseline on the EMDB dataset, which achieves a 1.94 ATE compared to a 2.33

Table 5: **Effectiveness of JOSH on dense scene reconstruction**. We evaluate on the EMDB (Kaufmann et al., 2023), SLOPER4D (Dai et al., 2023), and RICH (Huang et al., 2022) datasets and compare different variants of JOSH with methods using the same scene reconstruction initializations.

| | EMDB | SLOPER4D | | | | RICH |
|---|---|---|---|---|---|---|
| Methods | ATE ↓ | ATE ↓ | Abs Rel ↓ | $\delta<1.25$ ↑ | CD ↓ | CD ↓ |
| DROID-SLAM | 8.93 | 30.00 | 0.33 | 0.38 | 21.33 | 1.81 |
| DROID-SLAM +$JOSH_1$ | **5.57** | **17.18** | **0.25** | **0.57** | **16.84** | **1.57** |
| MonST3R | 2.33 | 15.06 | **0.21** | 0.74 | 13.28 | 1.70 |
| MonsT3R +$JOSH_2$ | **1.94** | **14.53** | 0.22 | **0.75** | **9.09** | **1.66** |
| MASt3R | 1.37 | 22.47 | 0.49 | 0.08 | 12.35 | 1.70 |
| MASt3R +$JOSH_3$ | **1.02** | **3.21** | **0.17** | **0.87** | **5.31** | **1.35** |

Table 6: **Global human motion estimation results on EMDB**.

| | EMDB | | | | |
|---|---|---|---|---|---|
| Methods | WA-MPJPE ↓ | W-MPJPE ↓ | RTE% ↓ | Jitter ↓ | FS ↓ |
| TRACE (Sun et al., 2023) | 529.0 | 1702.3 | 17.7 | 2987.6 | 370.7 |
| GLAMR (Yuan et al., 2022) | 280.8 | 726.6 | 11.4 | 46.3 | 20.7 |
| SLAHMR (Ye et al., 2023) | 326.9 | 776.1 | 10.2 | 31.3 | 14.5 |
| WHAM (Shin et al., 2024) | 131.1 | 335.3 | 4.1 | 22.5 | 4.4 |
| WHAC (Yin et al., 2024) | 142.2 | 389.4 | - | - | - |
| OfCaM (Yang et al., 2024) | 108.2 | 317.9 | 2.2 | - | - |
| COIN (Li et al., 2024) | 152.8 | 407.3 | 3.5 | - | - |
| GVHMR (Shen et al., 2024) | 109.1 | 274.9 | 1.9 | 16.7 | **3.5** |
| TRAM (Wang et al., 2024b) | 76.4 | 222.4 | 1.4 | 18.5 | 23.4 |
| PromptHMR-Vid (Wang et al., 2025b) | 71.0 | 216.5 | **1.3** | 16.3 | **3.5** |
| $JOSH_3$ (ours) | **68.9** | **174.7** | **1.3** | **9.1** | 12.1 |

ATE. This further suggests a strong correlation between the global human motion estimation and dense scene reconstruction, and the fact that performing joint optimization is necessary.

# E    MORE COMPARISON ON THE EMDB DATASET

As EMDB (Kaufmann et al., 2023) is the most widely used dataset for evaluating global human motion estimation, we perform an extensive comparison with other methods in Tab. 6. The results show that $JOSH_3$ sets up a new state-of-the-art on human motion estimation accuracy with a 68.9 WA-MPJPE and a 174.7 W-MPJPE. It also achieves decent motion quality with a 9.1 jittering and a 12.1 foot skating, which are comparable to other methods.

# F    ROBUSTNESS TO NOISY INITIALIZATIONS

We conducted additional experiments on the robustness of JOSH to noisy initializations. Specifically, we randomly perturb 20% of the local HMR, scene depth, and contact labels. For depth labels, we add a random unit Gaussian noise to the depth values. For local HMR labels, we add a random unit Gaussian noise to the local human translation. For contact labels, we randomly replace a contact joint with another one. The results are shown in Tab. 7.

We observe that JOSH is robust to contact-label noise, as it leverages both the dynamic mask and the monocular depth estimates to filter incorrect contact correspondences as discussed in Sec. 3.3, Contact Scene Loss. In addition, our contact modeling employs a robust loss similar to Mast3R-SfM (Duisterhof et al., 2024), further reducing the impact of outliers. For depth-label noise, the degradation is more pronounced in scene reconstruction than in global human motion. Conversely, local HMR label noise affects global human motion and physical plausibility more severely than scene reconstruction. Despite these perturbations, JOSH consistently outperforms SyncHMR (Zhao et al., 2024) across most evaluation metrics under all three noise conditions. These results demonstrate that JOSH maintains robustness from noisy inputs, which is crucial for 4D human–scene reconstruction in real-world settings.

Table 7: **Robustness analysis of JOSH on SLOPER4D**.

| Methods | SLOPER4D | | | | | | | | | |
|---|---|---|---|---|---|---|---|---|---|---|
| | WA-MPJPE ↓ | W-MPJPE ↓ | RTE ↓ | ATE ↓ | Abs Rel ↓ | $\delta < 1.25$ ↑ | CD ↓ | Jitter ↓ | FS ↓ | FFR ↓ |
| JOSH$_3$ | **120.0** | **438.3** | **1.7** | **3.21** | **0.16** | **0.87** | 5.31 | 7.1 | **28.2** | **2.9** |
| JOSH$_3$ (+noisy depth) | 245.5 | 865.9 | 4.9 | 12.07 | 0.41 | 0.63 | 6.24 | 7.8 | 39.1 | 3.0 |
| JOSH$_3$ (+noisy contact) | 121.6 | 502.6 | 1.8 | 4.06 | 0.32 | 0.84 | 5.35 | **6.4** | 32.5 | 3.8 |
| JOSH$_3$ (+noisy human) | 407.5 | 910.6 | 1.8 | 3.81 | 0.32 | 0.84 | **5.19** | 19.3 | 119.3 | 19.4 |
| SynCHMR$^\star$ | 233.2 | 1125.4 | 4.5 | 17.47 | 0.25 | 0.55 | 17.76 | 67.4 | 123.9 | 9.0 |

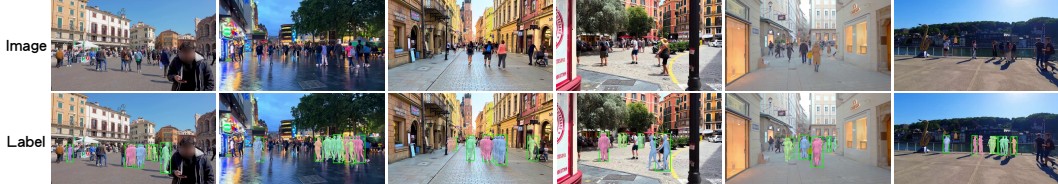

Figure 9: **Web videos annotated by JOSH**. The first row shows the initial frame of the sequence with pedestrians walking in urban scenes. The second row shows pseudo-labels of the global human motion predicted by JOSH by projecting the future motion to the initial frame.

# G  LABELING WEB VIDEOS WITH JOSH

We collect 20 hours of web videos of pedestrians walking in cities worldwide posted by content creators on YouTube (Sczepansky, 2024) with diverse pedestrian movements and scene backgrounds. All videos have a Creative Commons license and we remove all personally identifiable information when processing them. We split each video into 5-second clips (150 frames) and collected 3175 video clips. To annotate the global human motion, we first use VitDet (Li et al., 2022) to detect and DEVA (Cheng et al., 2023) to track the pedestrians and run the JOSH$_3$ variant to optimize the multi-human motion of all pedestrians and the scene background in each video clip. In total, we collect about 460,000 frames of global human motion pseudo-labels. Note that while it is possible to collect more labeled data for training the end-to-end human trajectory model JOSH3R, the current data scale and the diversity of scene backgrounds and human subjects are already much larger than existing real-world global human motion datasets EMDB2 (Kaufmann et al., 2023) and SLOPER4D (Dai et al., 2023) with only about 100,000 annotated frames and a few subjects and scenes. This further shows that the lack of diversity of existing datasets can limit the learning of end-to-end global human motion models from video inputs and the high potential of learning such models with large-scale pseudo-labels from web data. Some visualizations of the pseudo labels are depicted in Fig. 9, showing the high diversity of pedestrian movements and scene context from web videos.

# H  ARCHITECTURE OF JOSH3R

A more detailed architecture of JOSH3R is shown in Fig. 10. We keep the model design simple as a baseline for an end-to-end global human motion model. We build on top of the siamese network design of MASt3R (Leroy et al., 2024). JOSH3R first takes the feature maps $\boldsymbol{S}^i, \boldsymbol{S}^j$ after the frozen MASt3R encoder and decoder layers and applies ROI pooling with the bounding boxes from ViTDet to obtain the features of the scene within the spatial context of the human. We use an additional feature regressor that learns to extract relevant information from these features into human tokens $\boldsymbol{H}^i, \boldsymbol{H}^j$. Then, we concatenate $\boldsymbol{S}^i$ with $\boldsymbol{H}^i$ and $\boldsymbol{S}^j$ with $\boldsymbol{H}^j$ to form a complete set of tokens that encapsulates both human and scene information. These tokens are fed into two separate decoders, which contain cross attention modules where human and scene information can interact. The remaining network consists of several prediction heads. The decoded human tokens $\boldsymbol{H}^i, \boldsymbol{H}^j$ are each passed into a human prediction head that outputs the human's local translation $\boldsymbol{T}^i$ and $\boldsymbol{T}^j$ in their respective camera coordinates. The decoded scene tokens $\boldsymbol{S}^i, \boldsymbol{S}^j$ are max-pooled then passed through a scene prediction head that outputs $\boldsymbol{\sigma}^i, \boldsymbol{\sigma}^j$, which can then be used to scale the original MASt3R pointmap predictions as such: $\sigma^i \cdot \boldsymbol{X}_{11}, \sigma^j \cdot \boldsymbol{X}_{21}$. We reason that current scene reconstruction methods primarily lack in its scale and hence it suffices to predict the factor at which to scale the scene. Finally, the two human

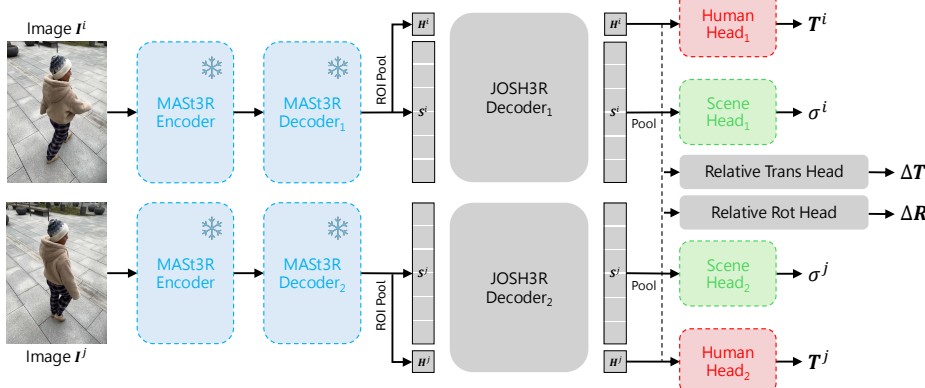

Figure 10: **JOSH3R joint human-scene prediction architecture**.

tokens and two pooled scene tokens are concatenated and inputted into two heads, which together produce relative rotation and translation $\Delta T, \Delta R$.

We train JOSH3R for 50 epochs on in-the-wild videos annotated by JOSH. The learning rate is set to 1e-4 with a batch size of 32 on 8 Nvidia RTX A6000 GPUs. We sample image pairs with temporal intervals between 1-10 frames during training and also apply data augmentations, including random center cropping and color jittering. During inference, the image pairs are sampled every 0.2s, and we also interpolate the human trajectories in between. We combine the global human trajectories from JOSH3R with the predicted local human poses from HMR2.0 (Goel et al., 2023) to get global human motion results.

