# OpenReview forum: "Joint Optimization for 4D Human-Scene Reconstruction in the Wild"
_ICLR.cc/2026/Conference — ICLR 2026 Poster_

### Official Review · Reviewer_rgPA · 2025-10-30

**Soundness:** 3
**Presentation:** 3
**Contribution:** 3
**Rating:** 6
**Confidence:** 5

**Summary:**

This paper proposes JOSH, a unified optimization framework for 4D human scene reconstruction in the wild from monocular video. This method introduces a human-scene contact loss, which enforces physical plausibility and consistency between the human scene and the surrounding environment. Several experiments on the SLOPER4D, EMDB, and RICH datasets demonstrate that JOSH significantly improves global human motion estimation, dense scene reconstruction, and physical plausibility metrics compared to previous methods (e.g., SynCHMR, WHAM, and TRAM). Another interesting network is JOSH3R, an end-to-end model that generates scalable pseudo-labels for large-scale web video training.

**Strengths:**

1. Josh appears to have a lot of competing work these days, but in terms of timeline, Josh is indeed the first to simultaneously optimize camera pose, human motion, and scene geometry using a joint optimization framework.
2. Josh3R is a good model that demonstrates the feasibility of automatically labeling real web videos for large-scale model training.
3. Josh's extensive experiments on multiple datasets (SLOPER4D, EMDB, and RICH) validate the effectiveness of his approach, which I highly recommend.

**Weaknesses:**

1. The overall visualization and experimental results (WA-MPJPE, MPJPE) are acceptable. However, I observed that even with the foot contact loss, there remains some penetration in the visualization (the 41-second mark in the supplementary video). Additionally, given that your work builds on the Rich model, why not consider adding hand penetration constraints? For instance, the first two visualizations in the supplementary video exhibit significant hand penetration.
2. The method relies heavily on the initialization model. While monocular camera scene reconstruction methods are advancing rapidly, JOSH still adopts MASt3R. Also, key experiments comparing with this year’s methods (e.g., VGGT, CUT3R) are missing. Including these experiments would better validate the work’s performance.
3. Although optimization-based methods are effective, they are computationally costlier than purely learning-based methods. It would be valuable to see experiments on time complexity to evaluate JOSH’s computational performance.
4. Theoretically, JOSH3R is capable of real-time performance. I would appreciate seeing results for JOSH3R—both qualitative visualizations and quantitative metrics.

**Questions:**

Please clarify why penetration remains despite foot contact loss and discuss the potential of adding hand penetration constraints. Provide results with other new monocular camera scene reconstruction methods. Add comparisons with recent methods (e.g., VGGT, CUT3R). Provide JOSH’s time complexity data and JOSH3R’s qualitative/quantitative results to validate real-time capability.

---

> ### Author Response · Authors · 2025-11-21
>
> Dear reviewer QBLa,
> Thank you for your valuable and helpful feedback. We sincerely appreciate the supportive comments, and you “highly recommend” JOSH’s extensive experiments.
>
> **Q1. Penetration Issue**.
>
> **A1**. Penetration cannot be completely eliminated because the entire JOSH framework is based on **joint optimization**. Although the contact loss Lc1 ​ encourages minimizing penetration, it must be balanced with several other objectives in the optimization, such as scene reconstruction and motion plausibility. As a result, the optimizer cannot always drive penetration to zero, especially with noisy inputs.
>
> Note that the contact loss is already applied to all human body parts, not just foot contact. The remaining hand-penetration primarily arises from limitations of the contact prediction model: while it can reliably predict foot contact, its hand-contact predictions are often very noisy. Please refer to *contact_vis.mov* in the supplementary material for a visualization of these noisy contact labels. We also observe that the hand contact is not correctly predicted when the person’s hands are touching the ground in the RICH example.
>
> **Q2. Comparison with Recent Methods**
>
> **A2**. While many feed-forward 4D reconstruction methods released after MASt3R focus on general dynamic point cloud reconstruction, they are not designed to recover articulated human meshes. Moreover, these approaches typically cannot reconstruct scenes at a metric scale due to the absence of human-based constraints. Their feed-forward nature also makes post-optimization difficult, especially in detailed contact modelling. JOSH is complementary to these methods: it can be integrated with them to provide stronger depth initialization and to refine the reconstruction through joint optimization.
>
> As **$\pi^3$** [1] is currently the strongest feed-forward 4D reconstruction method (outperforming VGGT [2] and CUT3R [3]), we further integrate $\pi^3$ into our pipeline and evaluate two variants: JOSH ($\pi^3$ + VIMO) and $\pi^3$ scaled using a metric-depth prior (ZoeDepth). We report results on SLOPER4D below.
>
> | SLOPER4D               | WA-MPJPE ↓ | W-MPJPE ↓ | RTE ↓ | ATE ↓ | Abs Rel ↓ | δ<1.25 ↑ | Jitter ↓ | FS ↓ | FFR ↓  |
> |----------------------|------------|-----------|-------|-------|-----------|----------|-----------|-------|--------|
> | **JOSH (MASt3R + VIMO)** | **120.0**     | **438.3**     | 1.7   | **3.21**  | **0.16**      | 0.87     | **7.1**       | **28.2**  | 2.9    |
> | **JOSH ($\pi^3$ + VIMO)**    | 122.1     | 483.9     | **1.0**   | 3.66  | 0.32      | **0.91**     | **7.1**       | 36.9  | **0.0**    |
> | **$\pi^3$ + ZoeDepth**       | 258.5     | 3831.3    | 18.5  | 63.49 | 1.68      | 0.22     | 143.6     | 105.1 | 0.54   |
>
> The results show that replacing MASt3R with $ \pi^3$ in JOSH also yields strong performance, thanks to the high-quality depth initialization provided by $\pi^3$. In contrast, the $\pi^3$ + ZoeDepth baseline fails dramatically across all metrics, highlighting that simply scaling feed-forward 4D reconstruction with monocular depth priors is insufficient for accurate and consistent 4D human–scene reconstruction. Thanks to the general framework of JOSH, it can benefit from stronger 4D reconstruction methods in the future.
>
> [1] Wang et al. $\pi^3$: Permutation-Equivariant Visual Geometry Learning. Arxiv 2025.
>
> [2] Wang et al. VGGT: Visual Geometry Grounded Transformer. CVPR 2025.
>
> [3] Wang et al. Continuous 3D Perception Model with Persistent State. CVPR 2025.

---

> ### Author Response · Authors · 2025-11-21
>
> **Q3. JOSH’s time complexity and Results for JOSH3R**
>
> **A3**. Thanks for the advice to analyze the time complexity and include more results of JOSH3R. We report the efficiency of JOSH and the efficiency and performance of JOSH3R as follows.
>
> | Methods  | FPS ↓ | EMDB WA-MPJPE ↓ | EMDB W-MPJPE ↓ | EMDB RTE% ↓ | SLOPER4D WA-MPJPE ↓ | SLOPER4D W-MPJPE ↓ | SLOPER4D RTE% ↓ | RICH WA-MPJPE ↓ | RICH W-MPJPE ↓ | RICH RTE% ↓ |
> |----------|--------|------------------|------------------|--------------|-----------------------|----------------------|-------------------|-------------------|------------------|---------------|
> | **SLAHMR** | 0.1    | 326.9            | 776.1            | 10.2         | 706.9                 | 3401.0               | 12.6              | 132.2             | 237.1            | 6.4           |
> | **WHAM**   | 4.4    | 131.1            | 335.3            | 4.1          | 297.7                 | 1272.3               | 10.5              | 108.4             | 196.1            | 4.5           |
> | **TRAM**   | 1.1    | 76.4             | 222.4            | 1.4          | 215.1                 | 1285.3               | 3.0               | 127.8             | 238.0            | 6.0           |
> | **JOSH_3** | 0.8    | **68.9**             | **174.7**            | **1.3**          |  **120.0**                 |  **438.3**                | **1.8**               | **102.6**             | **184.3**            |  **3.4**           |
> | **JOSH3R** | **15.4**   | 220.0            | 661.7            | 13.1         | 296.3                 | 1148.4               | 10.4              | 190.4             | 334.9            | 6.3           |
>
> Since JOSH3R takes a data-driven approach, it performs best on the SLOPER4D dataset, which is most similar to pedestrian street walking videos from JOSH labels. We can observe that while JOSH achieves the best accuracy, JOSH3R is significantly faster than JOSH, with the best efficiency of 15.4 FPS vs. 0.8 FPS, allowing real-time inference.
>
> As suggested by reviewer “u1bR”, we have also updated the manuscript to include JOSH3R as one of our contributions, as marked by the blue text. We would like to stress that the main contribution of our paper is the joint optimization framework of JOSH, and that JOSH3R is intended as a preliminary study of learning-based global human trajectory prediction trained on noisy in-the-wild labels. Although JOSH3R's current performance is only comparable to that of traditional methods, its results can still provide valuable insights to the community on training efficient end-to-end global human motion estimation models with pseudo-labels and powerful visual-geometry backbones. Some qualitative results of JOSH3R can be found in the supplementary (*JOSH3R.pdf*).

---

### Official Review · Reviewer_QBLa · 2025-10-31

**Soundness:** 3
**Presentation:** 3
**Contribution:** 2
**Rating:** 4
**Confidence:** 4

**Summary:**

This paper proposes a single-stage optimization framework that jointly estimates camera intrinsics/extrinsics, global multi-human motion (SMPL), and dense scene geometry from a in-the-wild monocular video. The key novelty is to explicitly enforce human–scene contact through two losses: (i) a contact-scene loss that pulls predicted human contact vertices toward matched scene points, and (ii) a contact-static loss that discourages contact sliding across time. The method also optimizes focal length jointly with root depth to correct metric-scale drift common in monocular setups. On SLOPER4D/EMDB/RICH, JOSH variants (with different off-the-shelf initializers) improve human trajectory errors, scene Chamfer distance, and physical plausibility (jitter, foot sliding) over strong baselines and a re-implementation of SynCHMR. They further show that pseudo-labels produced by JOSH enable scalable training of an end-to-end model that outperforms training on limited GT.

**Strengths:**

1. Clean contact formulation. Explicit vertex–scene distance with visibility + depth-prior gating + a temporal static contact term that mathematically targets sliding, which is practical and easy to implement without a simulator.
2. Focal-length optimization tied to root depth, addressing metric-scale failures when intrinsics are unknown, high leverage for web videos.
3. General wrapper. Boosts multiple scene/human initializers; evaluation spans human, scene, and physics plausibility metrics.

**Weaknesses:**

1. No force/stability reasoning. JOSH’s contacts are geometric (distance/static) without force, friction cone; expect residual artifacts under occlusion or weak depth priors (e.g., hand-on-sofa, compliant supports).
2. Contact detection & masking sensitivity. JOSH assumes reliable contact vertices and segmentation; occlusion/noisy masks may yield wrong correspondences, and the method’s robustness to such errors isn’t deeply quantified.
3. Dynamic scenes / non-static supports. geometric matching in JOSH presumes a static background for scene correspondences; cannot handle transient contacts and near-contacts through stability/dynamics.

**Questions:**

1. Evaluate foot-skate error, contact F1, and COM-support margin like PPR do.
2. Compare results with fixed f versus optimized f to show how intrinsics optimization resolves scale drift relative to SynCHMR.
3. Include contact visulizations with and without contact modeling.
4. Randomly perturb 20 % of BSTRO contact labels or depth priors; measure degradation in WA-MPJPE and sliding metrics to show robustness of Lc1/Lc2.
5. Provide inter-person collision or contact consistency metrics to substantiate multi-human capability.

I will reconsider my recommendation if the authors could kindly address these points.

---

> ### Author Response · Authors · 2025-11-21
>
> Dear reviewer VPyC,
> Thank you for your thorough and insightful comments. We sincerely appreciate the supportive feedback that our paper has a “clean contact formulation" and is a "general wrapper." We address your questions below.
>
> **Q1. Physical Metrics**
>
> **A1**. Following PPR [1], we report foot skating and contact F1, and compare it to the baseline SyncHMR.
> | SLOPER4D   | Foot Skating (cm) ↓ | Contact F1 (%) ↑ |
> |-----------|----------------------|-------------------|
> | **JOSH_3** | **2.8**                  | **35.7**              |
> | **SynCHMR** | 6.7                  | 7.8               |
>
> JOSH improves upon the baseline SyncHMR by a large margin, showing the effectiveness of improving the physical plausibility of the human motion from joint optimization. As SMPL is a purely kinematic human mesh model without physical modeling, it is difficult to reliably evaluate physical quantities such as the center of mass. On the other hand, JOSH is designed to address key challenges in 4D human–scene reconstruction, including: 1). **obtaining an accurate global human motion trajectory**. 2). **enabling more consistent human–scene interactions**, and 3). **reconstructing a metric-scale scene environment**, which is largely overlooked by existing literature. We believe enhanced physical modeling is an important and largely orthogonal research direction and will explore it in future work.
>
> **Q2. Intrinsics Optimization**
>
> **A2**. First, the original SynCHMR does not support optimizing camera intrinsics, as it is built on top of DROID-SLAM, which assumes fixed camera calibration. In contrast, JOSH supports intrinsic optimization thanks to its joint optimization framework and the coupling between human depth and focal length. We provide a comparison between fixed-f and optimized-f in the ablation study (Tab. 2), and the full comparison results are shown below.
>
> | SLOPER4D       | WA-MPJPE ↓ | W-MPJPE ↓ | RTE ↓ | ATE ↓ | Abs Rel ↓ | δ<1.25 ↑ | CD ↓ | Jitter ↓ | FS ↓ | FFR ↓ |
> |---------------|------------|-----------|-------|-------|-----------|----------|------|-----------|------|-------|
> | **Fixed f**    | 190.8      | 1220.7    | 5.5   | 19.89 | 0.60      | 0.10     | 11.5 | 8.0       | 71.3 | 15.2  |
> | **Optimized f**| 183.7      | 1053.4    | 4.6   | 16.91 | 0.47      | 0.31     | 10.4 | 7.9       | 68.3 | 6.8   |
>
> We can see that jointly optimizing the camera focal length f can lead to better reconstruction performance in all metrics, improving W-MPJPE from 1220.7 to 1053.5, ATE from 19.89 to 16.91, and FFR from 15.2\% to 6.8\%. These results show the importance of jointly optimizing camera intrinsics with JOSH for in-the-wild settings, where the ground truth is often unavailable.
>
> **Q3. Contact visualizations**
>
> **A3.** We have added new qualitative results for contact visualization (see *contact_vis.mov* in the supplementary material). As shown in the video, the predicted contact labels are inherently noisy—particularly for hand contacts. Despite this, JOSH is able to maintain 4D-consistent human–scene reconstruction and remains robust under such noisy contact input, while disabling contact modelling would lead to severe penetration and motion jittering.
>
> **Q4. Robustness of Lc1/Lc2**
>
> **A4**. Thanks for your suggestion to quantify JOSH’s robustness to input noise. Specifically, we randomly perturb 20% of the scene depth and contact labels.  For depth labels, we add a random unit Gaussian noise to the depth values. For contact labels, we randomly replace a contact joint with another one. The results are shown in the table below.
>
>
> | SLOPER4D                     | WA-MPJPE ↓ | W-MPJPE ↓ | Jitter ↓ | FS ↓ |
> |-----------------------------|------------|-----------|-----------|-------|
> | **JOSH_3**                  | 120.0      | 438.3     | 7.1       | 28.2  |
> | **JOSH_3 (+noisy depth)**   | 245.5      | 865.9     | 7.8       | 39.1  |
> | **JOSH_3 (+noisy contact)** | 121.6      | 502.6     | 6.4       | 32.5  |
> | **SynCHMR**                 | 233.2      | 1125.4    | 67.4      | 123.9 |
>
> We observe that JOSH is robust to contact-label noise, leveraging both the dynamic mask and monocular depth estimates to filter incorrect contact correspondences (Sec. 3.1, Contact Scene Loss). In addition, our contact loss Lc1/Lc2 employs a robust loss similar to Mast3R-SfM, further reducing the impact of outliers. For depth-label noise, the degradation is more pronounced in  WA-MPJPE than in foot sliding, so the resulting motion still has reasonable physical plausibility. As suggested by reviewer VPyC,  “The paper does this integration carefully, so that JOSH can benefit from the noisy predictions of the off-the-shelf systems.” Therefore, despite these perturbations, JOSH consistently outperforms SyncHMR across most evaluation metrics under all three noise conditions. These results demonstrate that JOSH maintains robustness from noisy inputs, which is crucial for 4D human–scene reconstruction in real-world settings.

---

> ### Author Response · Authors · 2025-11-21
>
> **Q5. Multi-human Capacity**
>
> **A5**. To evaluate the multi-human capability of JOSH, we further conduct experiments on in-the-wild videos with multiple annotated humans. Existing multi-human motion reconstruction methods typically benchmark on controlled datasets such as HI4D [2], captured from fixed cameras in studio settings. These works mainly report **Pair-PA-MPJPE** after Procrustes alignment. However, there is currently no standard benchmark or metric for evaluating global multi-human motion in unconstrained real-world videos. Therefore, we use a multi-person subset of the 3DPW [3] dataset, and assess performance using **Pair-WA-MPJPE_100** and **Pair-W-MPJPE_100**, where each pair of humans is aligned in world coordinates through a paired-global alignment procedure.
> | 3DPW (dynamic subset)  | Pair-WA-MPJPE₁₀₀ ↓ | Pair-W-MPJPE₁₀₀ ↓ |
> |-----------|----------------------|---------------------|
> | **JOSH_3** | **303.7**               | **547.2**              |
> | **TRAM**   | 339.6               | 1306.9             |
>
> We observe that JOSH_3 significantly outperforms the baseline TRAM in global multi-human motion accuracy as well. Despite these promising results, we acknowledge that robust multi-human modeling as well as benchmarks for global multi-human motion estimation—similar to physical modeling—is a complementary research direction to the main focus of JOSH. For instance, JOSH’s generality and flexibility make it possible to further incorporate multi-human contact constraints into the joint optimization framework, which could lead to more accurate human-human interaction modeling.
>
> [1] Yang et al. PPR: Physically Plausible Reconstruction from Monocular Videos. ICCV 2023.
>
> [2] Yin et al. Hi4D: 4D Instance Segmentation of Close Human Interaction. CVPR 2023.
>
> [3] Marcard et al. 3D Poses in the Wild Dataset. ECCV 2018.

---

### Official Review · Reviewer_VPyC · 2025-11-02

**Soundness:** 3
**Presentation:** 3
**Contribution:** 3
**Rating:** 6
**Confidence:** 4

**Summary:**

This paper proposes JOSH, an approach for reconstructing the 4D human motion and the 3D environment from a single video. The paper uses initialization information from off-the-shelf models and jointly optimizes the human motion and the environment. This is done by designing carefully the optimization objectives for this joint optimization. The complete system is benchmarked in different datasets and settings demonstrating strong quantitative performance.

**Strengths:**

- The proposed method achieves strong quantitative performance across the board. The paper provides comparisons with multiple baselines on a few benchmarks and JOSH outperform previous works in the majority of cases.
- Integrating signals for human contact is not easy, because these estimates are very noisy. The paper does this integration carefully, so that JOSH can benefit from the noisy predictions of the off the shelf systems.
- There is a helpful ablation that considers different aspects of the algorithm and shows the effect they have in the final result.
- The paper also introduces the JOSH3R model and trains it with pseudo-labels from running JOSH on in the wild videos. Although JOSH3R is not the main focus of the paper, it is a nice addition and shows the benefit of using JOSH for generating pseudo ground truth.

**Weaknesses:**

- The proposed approach is relatively straightforward for the most part. This type of optimizations are more traditional (e.g., Rempe et al, ICCV 2021 & Ye et al, CVPR 2023), so integrating better initial estimates from off-the-shelf models or refining the optimization objectives will often lead to better results.
- The use of human-scene contact relies on the contact being visible in the video. Such contact is often available in the simpler benchmark datasets used for evaluation but less common in in-the-wild videos. To be fair, this is something recognized by the paper as one of its weaknesses.
- The supplementary video does not provide a lot of results for in-the-wild videos. There are a few in the end, but they tend to be on the simpler side when it comes to the scene and/or the camera motion.

**Questions:**

- I appreciate the demonstration of using different off-the-shelf models for initializing the camera and human motion in Table 1 (e.g., VIMO, WHAM, etc), but it would be helpful to see some direct comparisons. I understand it might not be easy to do all the combinations, but how does VIMO+MonST3R compares with VIMO+MASt3R?
- I was underwhelmed from the demonstration on in-the-wild videos. It would be helpful to see results on more videos and more challenging ones. For example, how does the method performs on videos from the PoseTrack dataset?

---

> ### Author Response · Authors · 2025-11-21
>
> Dear reviewer VPyC,
>
> Thank you for your thoughtful and constructive feedback. We sincerely appreciate the supportive feedback that our work "achieves strong quantitative performance," “does this integration carefully,” and “has a helpful ablation.” We address your questions below.
>
> **Q1. VIMO + MonST3R vs. VIMO + MASt3R**
>
> **A1**. See the table below. We observe that JOSH (VIMO + MASt3R) outperforms JOSH (VIMO + MonST3R), while JOSH (VIMO + MonST3R) still surpasses JOSH(WHAM + MonST3R). This demonstrates the generality of JOSH and highlights that it can further benefit from both stronger depth and human initializations.
>
> | SLOPER4D                        | WA-MPJPE ↓ | W-MPJPE ↓ | RTE ↓ | ATE ↓ | Abs Rel ↓ | δ<1.25 ↑ | CD ↓ | Jitter ↓ | FS ↓ | FFR ↓ |
> |--------------------------------|------------|-----------|-------|-------|-----------|----------|------|-----------|------|-------|
> | **JOSH (VIMO + MASt3R)**       | 120.0      | 438.3     | 1.7   | 3.21  | 0.16      | 0.87     | 5.31 | 7.1       | 28.2 | 2.9   |
> | **JOSH (VIMO + MonST3R)**      | 148.9      | 585.7     | 4.2   | 14.09 | 0.21      | 0.75     | 7.92 | 6.2       | 32.4 | 2.8   |
> | **JOSH (WHAM + MonST3R)**      | 210.4      | 994.3     | 3.0   | 14.53 | 0.22      | 0.75     | 9.09 | 7.8       | 45.3 | 2.1   |
>
> **Q2. Qualitative results on PoseTrack.**
>
> **A2**. We have added new qualitative results on the challenging PoseTrack dataset (see *posetrack.mov* in the supplementary material), which contains complex multi-person videos. Note that 4D human–scene reconstruction from in-the-wild web videos remains challenging because the pipeline involves many interconnected components, and errors can easily propagate across stages. JOSH already incorporates several mechanisms to mitigate noise and outliers, and its primary goals are: 1). **obtaining acuurate global human motion trajectory**, 2). **enabling more consistent human–scene modeling**, and 3). **reconstructing a metric-scale scene** — aspects that we believe are crucial for 4D human–scene reconstruction yet largely overlooked by prior work.  We view many active research directions—such as improved local HMR, more robust detection and tracking, and better contact modeling and prediction—as complementary to our framework and will explore them in future work.

---

### Official Review · Reviewer_u1bR · 2025-11-04

**Soundness:** 3
**Presentation:** 2
**Contribution:** 2
**Rating:** 6
**Confidence:** 5

**Summary:**

- The paper tackles joint scene and human-mesh reconstruction from video, with a focus on modeling human–scene contacts.
- The proposed approach, JOSH, is an optimization-based method with two stages: (1) initialize point maps, correspondences, human meshes, and contact labels using off-the-shelf models; (2) jointly optimize the scene point cloud, camera parameters, and mesh parameters using priors and contact losses.
- The method introduces heuristics and losses—static-contact loss, scene-contact loss, and focal-length optimization—to accurately model the camera and humans in contact with the scene.
- The evaluations are done on multiple datasets: SLOPER4D, EMDB and RICH.
- Key Baselines: SLAHMR, WHAM, TRAM.
- The quantative results show that JOSH consistently outperforms the baselines across various scenarios.

**Strengths:**

- The paper is well written, organized and easy to follow. The ideas are presented clearly and the introduction section particularly is an insightful read.

- Joint human-scene modeling: The problem of jointly modeling scenes and humans is ambitious and forward-looking. The proposed approach, JOSH, is promising, shows strong in-the-wild results, and should encourage the community to design integrated human–scene methods.

- Simplicity of the approach: The core is simple—and that is its strength: initialize correctly and optimize reliably. This offers a promising path to obtain pseudo-3D (human+scene) supervision at scale from web videos. Although the supervision is noisy (compared to capture studios or synthetic), it remains to be seen how learning-based approaches (e.g., JOSH3R) generalize when trained at scale.

- Strong results: The qualitative results in the manuscript (and supplemental videos) clearly show the performance gap between JOSH and baselines such as TRAM and WHAM. The ability to model contacts over long durations without drift is impressive. Quantitatively, it outperforms both human- and scene-reconstruction baselines (MonST3R, MASt3R) across datasets.

**Weaknesses:**

- Limited Technical Novelty (without JOSH3R): I am suprised that JOSH3R's details (the learning based module built on top of the data collected using JOSH) is relegated to the appendix. Without JOSH3R as a core contribution, JOSH remains an optimization-based method for joint human–scene reconstruction; it can at best provide pseudo labels (although viable at scale). Standalone, it offers limited new technical insight. The optimization procedures largely mirror those in Hi4D (CVPR 2023), Ego-Exo4D (CVPR 2024), SLAMHR (CVPR 2023) with the key difference that the scene is included in the optimization. I acknowledge that incorporating the scene is nontrivial and requires precise execution at scale; however, this does not substantially increase the technical novelty. I strongly encourage including JOSH3R in the main paper narrative.

- Reliance on off-the-shelf initializations: The method acknowledges it can “suffer from bad initializations.” In practice, it stitches VIMO for humans and MASt3R for scenes, so failures propagate into JOSH. The optimization terms can refine but cannot recover from catastrophic errors.

- Comparison with Human only Methods on EMDB: The paper evaluates global human motion on EMDB, while other human/scene metrics are reported on SLOPER4D (less common in HMR). For a fair comparison to human-only methods, please report MPJPE (not world-MPJPE) on EMDB1 (see Table 3 of EMDB) to assess JOSH’s pose-alignment performance post-optimization against state-of-the-art human-only baselines.

**Questions:**

My questions are primarily centered around the weaknesses mentioned above:

1. Including JOSH3R in the main narrative. If not, justifications for novelty without it.
2. Robustness to poor initializations.
3. Pose-alignment comparison on EMDB-1.

---

> ### Author Response · Authors · 2025-11-21
>
> Dear Reviewer u1bR,
>
> Thank you for your constructive feedback. We appreciate your positive comments and supportive feedback, especially when you find our work “should encourage the community to design integrated human-scene approaches”. We address your questions below.
>
> **Q1. Limited technical novelty (without JOSH3R)**
>
> **A1**. Thanks for the advice to include JOSH3R in the main paper narrative. We have updated the manuscript to include JOSH3R as one of our contributions, as marked by the blue text. We would like to stress that the main contribution of our paper is the joint optimization framework of JOSH, and that JOSH3R is intended as a preliminary study of learning-based global human trajectory prediction trained on noisy in-the-wild labels. Although JOSH3R's current performance is only comparable to that of traditional methods, its results can still provide valuable insights to the community on training efficient end-to-end global human motion estimation models with pseudo-labels and powerful visual-geometry backbones. Quantitative results of JOSH3R has also been included in Tab.3 of the main paper and quantitative results of JOSH3R can be found in the supplementary (*JOSH3R.pdf*).
>
> **Q2. Robustness to poor initializations**
>
> **A2**. We conducted additional experiments on the robustness of JOSH to noisy initializations. Specifically, we randomly perturb 20% of the local HMR, scene depth, and contact labels.  For depth labels, we add a random unit Gaussian noise to the depth values. For local HMR labels, we add a random unit Gaussian noise to the local human translation. For contact labels, we randomly replace a contact joint with another one. The results are shown in the table below.
>
> | SLOPER4D                 |  WA-MPJPE ↓ | W-MPJPE ↓ | RTE ↓ | ATE ↓ | Abs Rel ↓ | δ<1.25 ↑ | CD ↓ | Jitter ↓ | FS ↓ | FFR ↓ |
> |-------------------------|----------------------|---------------------|-------|-------|-----------|-----------|------|-----------|------|-------|
> | **JOSH_3**              | 120.0               | 438.3              | 1.7   | 3.21  | 0.16      | 0.87      | 5.31 | 7.1       | 28.2 | 2.9   |
> | **JOSH_3 (+noisy depth)**   | 245.5               | 865.9              | 4.9   | 12.07 | 0.41      | 0.63      | 6.24 | 7.8       | 39.1 | 3.0   |
> | **JOSH_3 (+noisy contact)** | 121.6               | 502.6              | 1.8   | 4.06  | 0.32      | 0.84      | 5.35 | 6.4       | 32.5 | 3.8   |
> | **JOSH_3 (+noisy human)**   | 407.5               | 910.6              | 1.8   | 3.81  | 0.32      | 0.84      | 5.19 | 19.3      | 119.3| 19.4  |
> | **SynCHMR**             | 233.2               | 1125.4             | 4.5   | 17.47 | 0.25      | 0.55      | 17.76| 67.4      | 123.9| 9.0   |
>
> We observe that JOSH is the most robust to contact-label noise, as it leverages both the dynamic mask and the monocular depth estimates to filter incorrect contact correspondences (Sec. 3.1, Contact Scene Loss). In addition, our contact modeling employs a robust loss similar to Mast3R-SfM, further reducing the impact of outliers. For depth-label noise, the degradation is more pronounced in scene reconstruction than in global human motion. Conversely, local HMR label noise affects global human motion and physical plausibility more severely than scene reconstruction. As suggested by reviewer VPyC,  “The paper does this integration carefully, so that JOSH can benefit from the noisy predictions of the off-the-shelf systems.” Therefore, despite these perturbations, JOSH consistently outperforms SyncHMR across most evaluation metrics under all three noise conditions. These results demonstrate that JOSH maintains robustness from noisy inputs, which is crucial for 4D human–scene reconstruction in real-world settings.
>
> **Q3. Results on EMDB1.**
>
> **A3**. See the table below. Note that JOSH achieves local HMR performance comparable to its initialization, since local HMR is primarily determined by the human in the input video instead of the scene context/ camera poses. Adding scene-level constraints therefore does not yield significant improvement for this metric. In contrast, the main strengths of JOSH lie in: 1). **obtaining accurate global human motion trajectory**, 2). **enabling more consistent human–scene modeling**, and 3). **reconstructing a metric-scale scene environment**. Improving local HMR is an orthogonal research direction, and JOSH can benefit from advances in recent HMR methods with better initializations.
> | EMDB1        | PA-MPJPE ↓ | MPJPE ↓ | PVE ↓ | ACCEL ↓ |
> |----------------|------------|---------|-------|----------|
> | **VIMO**       | 46.6       | 76.6    | 89.5  | 6.4      |
> | **VIMO + JOSH**| 46.5       | 75.8    | 88.5  | 5.4      |
> | **WHAM**       | 50.4       | 79.7    | 94.4  | 5.3      |
> | **WHAM + JOSH**| 50.3       | 78.6    | 93.1  | 5.6      |

---

> ### Comment · Reviewer_u1bR · 2025-11-27
>
> Thank you for the rebuttal, my concerns are addressed. I would like to keep my rating.

---

### Comment · Area_Chair_mRjE · 2025-11-24

Dear Reviewer,

Thanks for taking the time to review this work. The authors have responded to your reviews. Can you please have a look at the rebuttal and discuss with the authors?

Best Regards,

AC

---

### Author Response · Authors · 2025-12-01
**Author-Reviewer Discussion Summary**

Dear ACs,

Thank you for taking the time to review our paper. We would like to summarize our discussion with the reviewers as follows.

The reviewers acknowledged the strong performance and extensive experiments presented in JOSH, while expressing concerns regarding its robustness to poor initializations. In response, we conducted additional experiments evaluating JOSH’s performance under various input noises. The results show that JOSH remains robust, particularly under noisy contact labels.

Several reviewers requested further demonstration of JOSH’s capabilities in local HMR, physical modeling, and multi-human modeling. In the rebuttal, we provide additional results in these areas, showing strong performance relative to baseline methods. We also note that these directions are complementary and orthogonal to the primary goals of JOSH, and represent promising avenues for future work.

Finally, following the suggestions of Reviewers u1bR and rpGA, we have updated the manuscript to include JOSH3R in the main narrative with additional qualitative and quantitative results to enhance the technical novelty and provide further insights.

We have also conducted the following experiments during the rebuttal, which further demonstrate the universality of JOSH as a general framework.
1. Comparison of VIMO + MASt3R vs VIMO+MonST3R
2. Comparison to the state-of-the-art 4D reconstruction method $\pi^3$
3. Qualitative evaluations on PoseTrack and visualizations of contact labels

Due to the limited response window, only Reviewer u1bR replied to our initial rebuttal, stating that “my concerns are addressed.”

Best Regards,

Authors of Submission 3882

---

### Meta-Review · Area_Chair_CNXD · 2026-01-12

**Summary:**

This paper jointly estimates camera intrinsics/extrinsics, human-mesh, and dense scene geometry in a single-stage framework. With strong quantitative results across multiple datasets.  The rebuttal substantively addresses concerns about novelty, robustness, and evaluation scope, making this paper from a borderline case to acceptance.

**Reviewer Concerns:**

Addressed Concerns:

The authors have moved JOSH3R into the main narrative and expanded its presentation (addressing u1bR's primary concern). Reviewer u1bR confirmed their concerns were addressed.

New experiments systematically testing robustness to noisy depth, contact, and human motion initializations were provided.

Outstanding Concerns

Reviewers noted that penetration, especially for hands, is not fully resolved, and that the method does not incorporate physical force/stability reasoning.

**Reviewer Scores:**

Reviewer u1bR (Initial 6): Explicitly stated "my concerns are addressed" and wished to keep their rating.

Reviewer VPyC (Initial 6): Their specific requests for backbone comparisons and in-the-wild demos were thoroughly addressed, score likely 6-7.

Reviewer QBLa (Initial 4): Provide new experiments, metrics, and visualizations, score likely increase to 6.

Reviewer rgPA (Initial 6): Their concerns about comparisons with new methods and JOSH3R results were addressed, score likely 6-7.

---

### Decision · Program_Chairs · 2026-01-26

Accept (Poster)